# Saffold virus exploits integrin αvβ8 and sulfated glycosaminoglycans as cooperative attachment receptors for infection

Takako Okuwa [1,4], Toshiki Himeda [1,4] ✉, Kyousuke Kobayashi [2,4], Namiko Nomura[2], Kouichi Utani[1], Satoshi Koike [2], Akira Nakamura[3] & Masaya Higuchi [1] ✉

Saffold virus (SAFV), a member of the species *Cardiovirus saffoldi* within the *Picornaviridae* family, causes acute respiratory and gastrointestinal illnesses as well as hand, foot, and mouth disease. It is also suspected to be associated with neuronal disorders, such as encephalitis and meningitis, in severe cases. Despite its clinical significance, the virus-host interactions underlying SAFV pathogenicity remain largely unknown. Using a genome-wide CRISPR-Cas9 knockout screen, we identify the following receptors for SAFV infection: sulfated glycosaminoglycans (GAGs) and integrin αVβ8. Single knockouts of *SLC35B2*, an essential gene for sulfated GAG synthesis, or the integrin genes *ITGAV* or *ITGB8* partially reduce SAFV-3 and SAFV-2 susceptibility in HeLa cells, and a double knockout confers complete resistance. Furthermore, we demonstrate that SAFV-3 virions bind directly to sulfated GAGs and integrin αVβ8. Based on these findings, we propose a model of SAFV infection in which sulfated GAGs and integrin αVβ8 act through dual and cooperative pathways to facilitate viral entry.

Saffold virus (SAFV) belongs to the species *Cardiovirus saffoldi* (formerly *Cardiovirus* D) within the genus *Cardiovirus* of the family *Picornaviridae*. It is closely related to Theiler's murine encephalomyelitis virus (TMEV), which is classified under the species *Cardiovirus theileri* (formerly *Cardiovirus* B)[1–3]. In 2007, SAFV was first isolated from a stool sample of an eight-month-old infant with fever of unknown origin in the United States[4]. To date, 11 different SAFV genotypes have been identified, with SAFV-2 and SAFV-3 being highly prevalent in humans[5]. SAFV is primarily detected in pediatric patients with acute respiratory illness and gastroenteritis, but it has also been found in specimens from severe cases, including acute flaccid paralysis, aseptic meningitis, myocarditis, acute pancreatitis, and cerebellitis[5–10]. In addition, SAFV infection has been associated with hand, foot, and mouth disease (HFMD), where an increased frequency of severe nervous system manifestations has been reported[11]. SAFV coinfection exacerbates the severity of HFMD caused by enterovirus 71 infection[11]. However, the pathogenicity of SAFV, which causes mild to severe symptoms, remains poorly understood.

The capsid of SAFV is composed of 60 capsomers, each containing four subunits: VP1, VP2, VP3, and VP4. The fourth capsid protein, VP4, is located inside the capsid[12]. Surface-exposed capsid proteins bind to receptors on the target cells, thereby initiating the first step of infection. This virus-receptor interaction is followed by viral entry, uncoating, replication, assembly, and ultimately, the release of progeny viruses from infected cells. Thus, the receptors and their distribution are the most important factors in defining the host range and tissue tropism and play a crucial role in viral pathogenicity. However, the SAFV receptor has not been identified.

[1]Department of Microbiology, Kanazawa Medical University School of Medicine, Ishikawa, Japan. [2]Neurovirology Project, Department of Genome Medicine, Tokyo Metropolitan Institute of Medical Science, Tokyo, Japan. [3]Division of Immunology, Faculty of Medicine, Tohoku Medical and Pharmaceutical University, Miyagi, Japan. [4]These authors contributed equally: Takako Okuwa, Toshiki Himeda, Kyousuke Kobayashi. ✉e-mail: himeda@kanazawa-med.ac.jp; masahigu@kanazawa-med.ac.jp

To identify SAFV receptors, we employed genome-wide CRISPR-Cas9 knockout (KO) screens, a widely used method for identifying critical host factors for viral infection, including receptors[13–18], using HeLa-N cells, which are a HeLa subline highly susceptible to SAFV-3 infection[19].

We herein report the identification of sulfated glycosaminogly-cans (GAGs) and integrin αVβ8 as receptors for SAFV and demonstrate that these two pathways play dual and cooperative roles during SAFV infection.

## Results

### Identification of genes involved in sulfated GAG synthesis as host factors for efficient SAFV-3 infection

To identify host factors required for SAFV infection, we used the human CRISPR KO pooled library (GeCKO v2)[20] and HeLa-N cells, which are highly susceptible to SAFV infection[19], along with the JPN08-404 strain of SAFV-3. To strengthen the reliability of the screening, we conducted two independent screens using separately established HeLa-N KO cell libraries, applying two different selection pressures by varying the SAFV-3 exposure time. Under strong selection pressure, where cells were continuously exposed to the virus for four days, almost none of the cells survived infection and only a few candidate genes were identified. In contrast, under milder selection pressure, where cells were exposed to the virus for just one day, more cells survived and a larger number of candidates were obtained (see Source data for Fig. 1a). We then searched for common hits between the screens and identified only two candidate host factors involved in SAFV-3 infection: *PA2G4* (proliferation-associated 2G4), an important translation initiation factor for the type II internal ribosome entry site (IRES) of foot-and-mouth disease virus (FMDV)[21] and cardioviruses[22]; and *SLC35B2* (solute carrier family 35 member B2), which is responsible for transporting the sulfate donor 3-phosphoadenosine-5-phosphosulfate (Fig. 1a). Subsequent validation experiments using individual gene knockouts of *PA2G4* or *SLC35B2* demonstrated that disruption of these factors significantly reduced susceptibility to SAFV-3 (Supplementary Fig. 1a). These findings suggest that PA2G4 plays a key role in the translation of the SAFV polyprotein from the IRES, similar to other cardioviruses such as TMEV and encephalomyocarditis virus (EMCV)[22], and that SLC35B2 may be responsible for the sulfation of cell surface molecules involved in SAFV attachment and entry. Consequently, we focused subsequent analyses on SLC35B2 to identify the receptors for SAFV.

SLC35B2 not only plays a role in GAG sulfation but is also important for regulating overall protein tyrosine sulfation[16,23–25]. We noticed that several genes involved in heparan sulfate (HS) synthesis, such as *EXT1* and *EXT2* (exostosin-1 and -2), were uniquely identified as candidates in the second screen, which was conducted under mild selection pressure (see Source data of the 2nd screening for Fig. 1a). To determine whether sulfated GAGs or sulfated proteins are critical for SAFV-3 infection, we established *SLC35B2* KO and *EXT1* KO HeLa-N cell lines (referred to as HeLaN-ΔSLC and HeLaN-ΔEXT1, respectively). EXT1 is a representative glycosyltransferase required for the elongation of the HS chain. We confirmed that HS was absent on the surface of these cells and that frameshift mutations were introduced into the target regions (Fig. 1b and Supplementary Fig. 2). The susceptibility of these cells to SAFV-3 infection was examined by inoculating them with serially diluted viruses. The susceptibility of HeLaN-ΔSLC cells to SAFV-3 was reduced by 3 logs compared to that of wild-type (WT) cells; however, HeLaN-ΔSLC cells were still eradicated by high-titer viral inoculation (Fig. 1c, upper panel). A similar reduction in SAFV-3 susceptibility in HeLaN-ΔSLC cells was observed when they were infected with SAF/UnaG virus, a recombinant virus expressing the green fluorescent protein UnaG in infected cells[26] (Supplementary Fig. 3). In addition, the susceptibility of HeLaN-ΔEXT1 cells to SAFV-3 was reduced by 2 or 3 logs compared with that of WT cells (Fig. 1c, lower

panel). These results suggest that the decreased susceptibility observed in *SLC35B2* KO cells was primarily due to the loss of sulfated GAGs, particularly HS proteoglycans, whereas overall protein tyrosine sulfation, which also requires *SLC35B2*[16,23–25], played a negligible role. Therefore, in subsequent experiments, we employed *SLC35B2* KO cells as a strategy to eliminate sulfated GAGs. Next, we assessed the growth kinetics of SAFV-3 in the HeLaN-ΔSLC and WT HeLa-N cells. Consistent with the reduced susceptibility of HeLaN-ΔSLC cells, they also supported SAFV-3 infection and replication, albeit with an approximately 2-log lower efficiency than that of WT cells (Fig. 1d). These findings indicate that sulfated GAGs are required for efficient SAFV-3 infection but strongly suggest the presence of another major receptor that functions in cooperation with sulfated GAGs or independently of them.

### Identification of integrin αV and integrin β8 as critical factors for SAFV-3 infection

To identify the major receptors for SAFV-3 infection, other than sulfated GAGs, we conducted a second genome-wide CRISPR screening using HeLaN-ΔSLC cells with two SAFV-3 strains (JPN08-404 and JPN08-356). The top two most enriched candidate genes identified in both screens were *ITGAV* and *ITGB8*, which encode the integrin subunits αV and β8, respectively (Fig. 2a), and their products form the integrin αVβ8 heterodimer. These two genes were significantly enriched, prompting us to focus on *ITGAV* and *ITGB8* in further studies.

To verify the role of integrin αVβ8 in SAFV-3 infection, we knocked out *ITGAV* or *ITGB8* in WT and HeLaN-ΔSLC cells (resulting in HeLaN-ΔAV, HeLaN-ΔB8, HeLaN-ΔSLCΔAV, and HeLaN-ΔSLCΔB8 cells) and established KO clones in which the loss of target gene expression on the cell surface and frameshift mutations were confirmed (Fig. 2b and Supplementary Figs. 2 and 4a). It has been reported that integrin αVβ8 is expressed on the cell surface only as a heterodimer[27]. We confirmed that *ITGAV* KO led to a significant reduction in the surface expression of integrin β8 (Supplementary Fig. 5). Therefore, in this study, we used a single KO of each subunit to effectively deplete functional integrin αVβ8 heterodimer.

Next, we examined their susceptibility to SAFV-3 by inoculating cells with serially diluted viruses (Fig. 2c). Although HeLaN-ΔAV and HeLaN-ΔB8 cells were eradicated by high-titer viral inoculation, their susceptibility to SAFV-3 was 3-log lower than that of the WT. In contrast, HeLaN-ΔSLCΔAV and HeLaN-ΔSLCΔB8 cells were completely resistant to SAFV-3 even at the highest viral multiplicity of infection (MOI). Two additional clones from each KO cell line showed a similar susceptibility to SAFV-3 (Supplementary Fig. 4b). When infected with the SAF/UnaG virus, UnaG-positive cells were detected in HeLaN-ΔAV and HeLaN-ΔB8 cells, albeit at a lower frequency than in WT cells, whereas no UnaG-positive cells were observed in HeLaN-ΔSLCΔAV and HeLaN-ΔSLCΔB8 double KO cells (Supplementary Fig. 3). We then assessed the growth kinetics of SAFV-3 in the HeLaN-ΔB8 single-KO and HeLaN-ΔSLCΔB8 double KO cells. As shown in Fig. 2d, viral growth in HeLaN-ΔB8 cells was considerably reduced, exhibiting delayed replication and decreased titers; viral growth still occurred, similar to that observed in HeLaN-ΔSLC cells. In contrast, no viral growth was detected in the HeLaN-ΔSLCΔB8 cells (Fig. 2d). To confirm whether the resistance of these KO cells is specific to SAFV-3, we tested their susceptibility to other picornaviruses, including EMCV (genus *Cardiovirus*) and coxsackievirus B3 (genus *Enterovirus*), which utilize distinct receptors[28–30]. KO cells exhibited susceptibility comparable to that of WT cells, confirming that their resistance is specific to SAFV-3 (Supplementary Fig. 6). These results indicate that in HeLa-N cells, knocking out either *SLC35B2* or integrin alone resulted in partial susceptibility to SAFV-3. However, knocking out *ITGAV* or *ITGB8* in addition to *SLC35B2* resulted in a complete loss of susceptibility to SAFV-3. This suggests the existence of two pathways for SAFV-3 infection: a sulfated GAG-dependent pathway and an integrin αVβ8-dependent pathway.

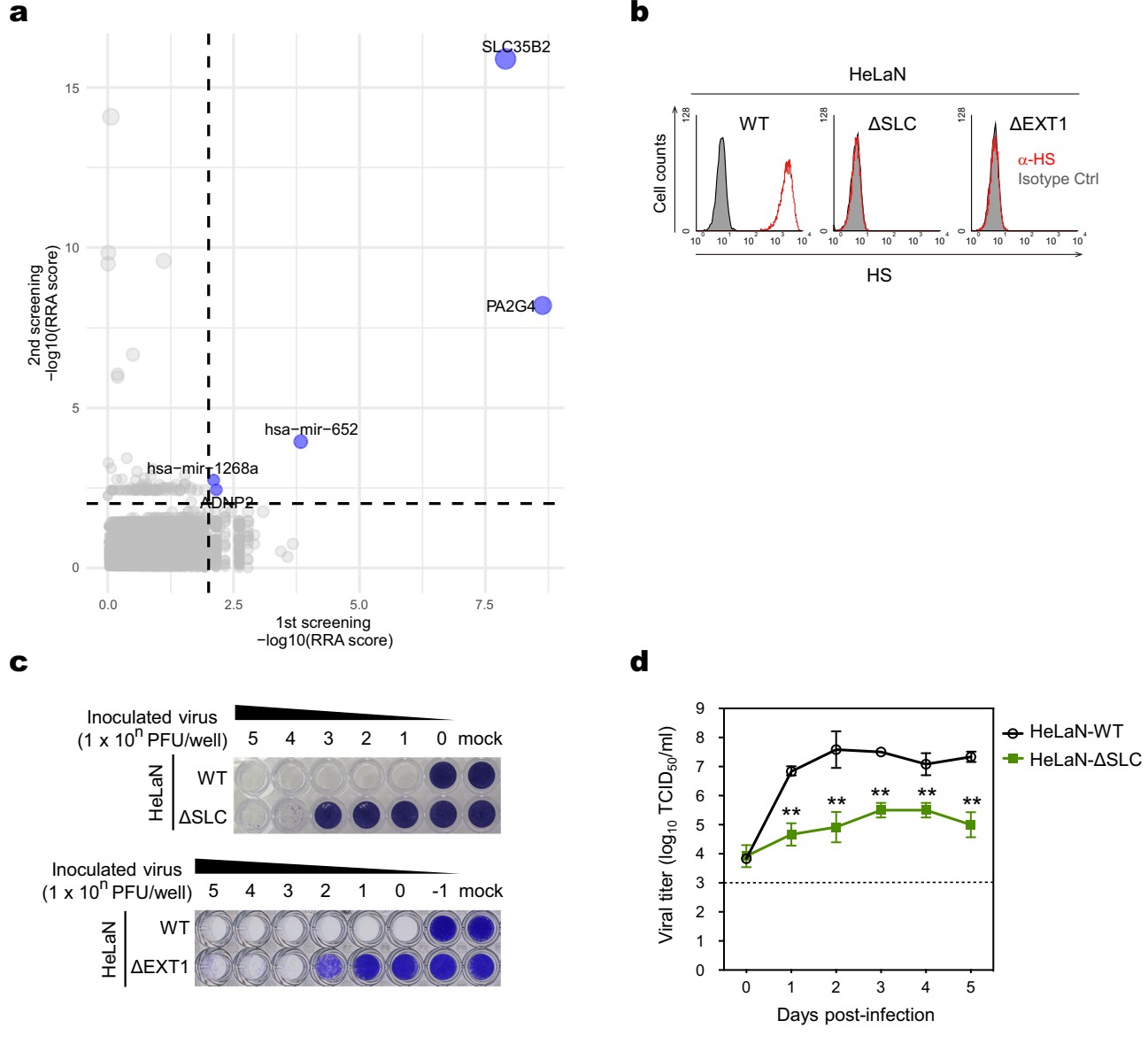

**Fig. 1 | *SLC35B2* KO reduces but does not completely inhibit SAFV-3 infection in HeLa-N cells. a** The graph displays the −log10-transformed robust ranking aggregation (RRA) scores of genes enriched following SAFV-3 infection in HeLa-N cells, analyzed using the MAGeCK software program. The *X*-axis represents data from the first CRISPR screen, while the *Y*-axis shows results from the second screen. The dotted line indicates the significance threshold of RRA = 0.01. Genes that met the criterion of RRA < 0.01 in both screens are highlighted in blue. The size of each dot reflects the combined enrichment across both screens, with larger dots indicating a greater sum of −log10 (RRA scores) from both experiments. **b** HS expression in HeLaN-ΔSLC cells. Non-permeabilized cells were stained with an anti-HS antibody and analyzed using flow cytometry. **c** WT, HeLaN-ΔSLC, and HeLaN-ΔEXT1 cells were infected with tenfold serial dilutions of SAFV-3, and viable cells were stained with crystal violet to assess the infection levels. Images are representative of two independent experiments. **d** Multi-step growth kinetics of SAFV-3 in HeLaN-ΔSLC and HeLaN-WT cells. The cells were infected with SAFV-3 and incubated for up to 5 days. Data are presented as mean viral titers with standard deviation (s.d.) (*n* = 3). Statistical significance was determined using the two-sided Welch's *t*-test. **,** *P* < 0.01. The dotted line indicates the limit of detection. Source data are provided as a Source Data file. Ctrl control.

In addition, among the other candidate genes identified in the second screen (see Source data for Fig. 2), we performed validation experiments on several factors reported to be localized to the membranes (Supplementary Fig. 1b). *PA2G4* was used as the positive control. However, all tested factors were confirmed as false positives.

## Other clinical isolates and genotypes also utilize both sulfated GAGs and integrin αVβ8

To determine whether the dual infection pathway mediated by sulfated GAGs and integrin αVβ8 is a common mechanism for SAFV infection, we analyzed the infection phenotypes of two additional clinical isolates of genotype 3 (JPN08-356 and 987/Niigata/2007) and

one clinical isolate of genotype 2 (1801-Yamagata-2009) (Fig. 3). The passage numbers of each clinical strain and cell lines used for their isolation are summarized in Supplementary Table 1. Susceptibility to the JPN08-356 strain was reduced by 2 logs in HeLaN-ΔSLC cells and by 2 logs in HeLaN-ΔB8 cells compared to control (WT) cells. For the 987/Niigata/2007 strain, susceptibility was 1 log lower in HeLaN-ΔSLC cells and 3 logs lower in HeLaN-ΔB8 cells than in WT cells. Similarly, the susceptibility to SAFV-2 strain 1801-Yamagata-2009 was reduced by 3 logs in HeLaN-ΔSLC cells and by 2 logs in HeLaN-ΔB8 cells compared to WT cells. Notably, none of the strains were able to infect the HeLaN-ΔSLCΔB8 cells. These results indicate that the infections of all strains examined in this study are mediated by

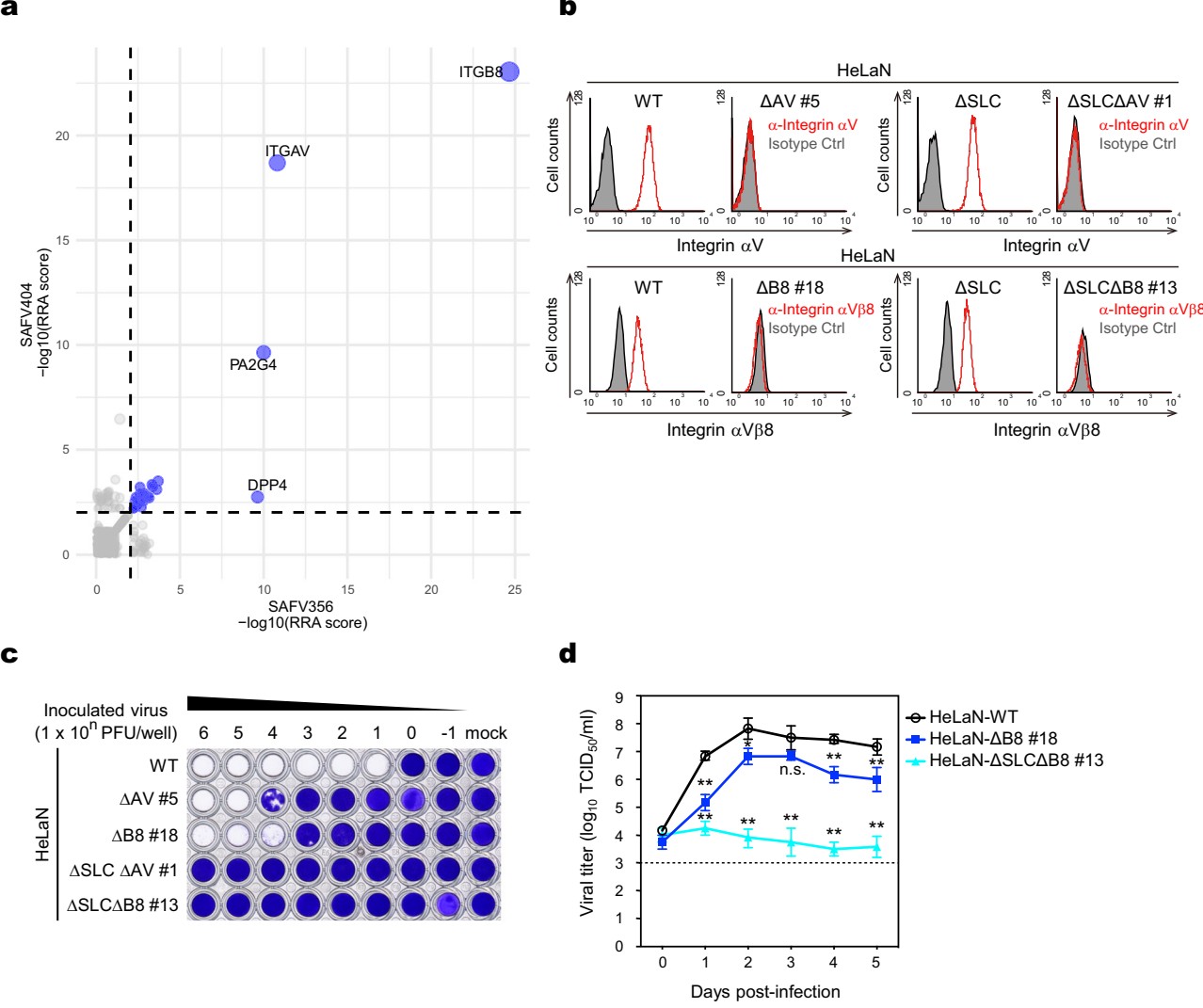

**Fig. 2 | *ITGAV* and *ITGB8* are other major factors for SAFV-3 infection. a** The graph displays the −log10-transformed RRA scores of genes enriched following infection with two SAFV-3 strains in HeLaN-ΔSLC cells, analyzed using the MAGeCK software. The *X*-axis represents data from the screen using the SAFV-3 JPN08-356 strain, whereas the *Y*-axis shows results from the screen using the SAFV-3 JPN08-404 strain. The dotted line indicates the significance threshold of RRA = 0.01. Genes that met the criterion of RRA < 0.01 in both screens are highlighted in blue. The size of each dot reflects the combined enrichment across both screens, with larger dots indicating a greater sum of −log10 (RRA scores) from both experiments. **b** Expression of human integrin αV and integrin β8 in HeLaN-WT, HeLaN-ΔAV, HeLaN-ΔSLCΔAV, HeLaN-ΔB8, and HeLaN-ΔSLCΔB8 cells. The cells were stained with anti-integrin αV or anti-integrin αVβ8 antibodies and analyzed by flow cytometry. **c** HeLaN-WT, HeLaN-ΔAV, HeLaN-ΔB8, HeLaN-ΔSLCΔAV, and HeLaN-ΔSLCΔB8 cells were infected with tenfold serial dilutions of SAFV-3 and viable cells were stained with crystal violet to assess infection levels. Images are representative of two independent experiments. **d** Multi-step growth kinetics of SAFV-3 in HeLaN-ΔB8, HeLaN-ΔSLCΔB8, and HeLaN-WT cells. The cells were infected with SAFV-3 and incubated for up to 5 days. Data are presented as mean viral titers with s.d. (*n* = 3). Statistical significance was determined using the two-sided Welch's *t*-test. **, *P* < 0.01, *, *P* < 0.05, n.s. not significant. The dotted line indicates the limit of detection. Source data are provided as a Source Data file.

both the sulfated GAGs and integrin pathways, at least in genotypes 2 and 3.

## Expression of integrin αVβ8 confers susceptibility to SAFV-3 in BHK-21 cells

BHK-21 cells, which are derived from hamsters, are resistant to SAFV-3 infection, even at a high MOI, but can produce progeny viruses when transfected with infectious SAFV-3 RNA[26]. This suggests that BHK-21 cells lack the key host factors involved in the attachment, internalization, or uncoating of SAFV-3. We hypothesized that if integrin αVβ8 is a missing factor, expressing it would confer SAFV-3 susceptibility to BHK-21 cells.

First, we analyzed the expression of integrins αV and β8 in BHK-21 cells. As antibodies to hamster integrins αV and β8 for flow cytometry were not available, we performed western blotting using anti-human integrin αV and anti-mouse integrin β8 antibodies, which are cross-reactive to hamster integrins (Fig. 4a). We detected endogenous hamster integrin αV as well as exogenously expressed human integrin αV, but could not detect endogenously expressed hamster integrin β8, demonstrating that BHK-21 cells express integrin αV but not β8 at detectable levels. Flow cytometry showed that HS was abundantly present on the surface of BHK-21 cells (Fig. 4b), indicating that sulfated GAGs alone were insufficient for SAFV-3 infection in these cells.

Next, we exogenously expressed human integrin αV and/or β8 in WT BHK-21 cells via lentiviral transduction (Fig. 4b) and assessed SAFV-3 susceptibility using SAF/UnaG virus (Fig. 4c). Expression of integrin β8 alone (BHK + human B8) conferred susceptibility to SAFV-3 in BHK-21 cells, whereas expression of integrin αV alone (BHK + human AV) did

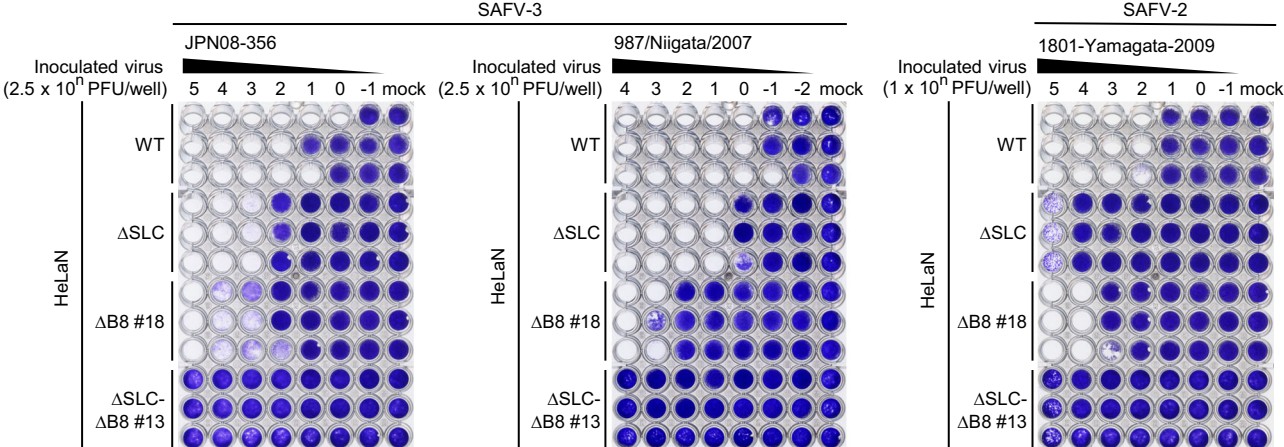

**Fig. 3 | SAFVs commonly utilize sulfated GAGs and integrin αVβ8 as dual receptors for infection.** Susceptibility to SAFV clinical isolates in KO HeLa-N cell lines. HeLaN-WT, HeLaN-ΔSLC, HeLaN-ΔB8, and HeLaN-ΔSLCΔB8 cells were infected with tenfold serial dilutions of the indicated SAFV clinical isolates and incubated for 5 days. Viable cells were stained with crystal violet. Images are representative of two independent experiments.

not. Co-expression of integrin αV and β8 (BHK + human AVB8) further increased SAFV-3 susceptibility in BHK-21 cells. Finally, we examined the growth kinetics of SAFV-3 in integrin-expressing BHK-21 and control cells. No viral growth was observed in control or BHK + human AV cells. However, drastic viral growth was observed in BHK + human B8 and BHK + human AVB8 cells (Fig. 4d). These results confirm that integrin αVβ8 is a critical factor in SAFV-3 infection.

Finally, we investigated whether the β8 subunits of human and rodent integrins are functionally interchangeable during SAFV-3 infection by assessing the susceptibility of BHK-21 cells expressing hamster or mouse integrin β8 using the SAF/UnaG virus (Fig. 4e, f). We observed a substantial increase in the number of UnaG-positive cells in both hamster and mouse integrin β8-expressing cells, comparable to that in cells expressing human integrin β8. These results suggest that the β8 subunits of human and rodent integrins are interchangeable during SAFV-3 infection.

### SAFV-3 specifically utilizes integrin αVβ8 among the integrin αV subfamily for infection

Integrin αV forms heterodimers with not only integrin β8 but also β1, β3, β5, and β6, constituting the integrin αV subfamily. Since some picornaviruses, such as FMDV, utilize multiple members of the integrin αV subfamily as receptors, we investigated whether SAFV-3 can utilize other integrin β subunits besides β8 for infection. We generated BHK-21 cells co-expressing integrin αV with each β subunit (β1, β3, β5, or β6) by lentiviral transduction (Fig. 5a) and assessed their susceptibility to SAFV-3 using SAF/UnaG virus. No obvious increase in the number of UnaG-positive cells was observed in cells expressing β subunits other than β8 (Fig. 5b). These results indicate that SAFV-3 specifically utilizes integrin αVβ8 among the αV subfamily members for infection.

### HS, as the representative sulfated GAG, and integrin αVβ8 interact directly with SAFV-3

We examined the direct binding of SAFV-3 to heparin, a structural analog of HS. Western blotting detected the SAFV-3 antigen in the sample pulled down by the heparin beads (Fig. 6a, left panel). We also examined the binding of SAFV-3 to integrin αVβ8 using a pull-down assay with Fc-tagged soluble integrin αVβ8 in both the presence and absence of $Ca^{2+}/Mg^{2+}$, because integrin αVβ8 undergoes a divalent cation-dependent conformational change to its active form, allowing ligand binding at the RGD (Arg-Gly-Asp)-binding site[27]. The SAFV-3 antigen was detected in a $Ca^{2+}/Mg^{2+}$-dependent manner in the sample pulled down by the integrin αVβ8 beads but not by the integrin αVβ3 beads (Fig. 6a, right panel). These data clearly demonstrated that

SAFV-3 can directly bind to sulfated GAGs and integrin αVβ8. Moreover, since SAFV-3 binding to integrin αVβ8 was $Ca^{2+}/Mg^{2+}$-dependent, the mode of interaction appears to be similar to that of typical integrin–ligand interactions involving RGD motifs.

To determine whether SAFV-3 binds to sulfated GAGs and integrin αVβ8 on the cell surface, we conducted a virus attachment assay using WT HeLa-N, HeLaN-ΔSLC, HeLaN-ΔB8, HeLaN-ΔSLCΔB8, and integrin αVβ8-overexpressing HeLaN-ΔSLC (HeLaN-ΔSLC + human AVB8) cells. These cells were incubated with SAFV-3 at 4 °C, and the viruses attached to the cells were quantified by quantitative reverse transcription polymerase chain reaction (RT-qPCR) (Fig. 6b). WT cells supported a high amount of virus attachment, whereas the amount of virus attached was significantly reduced in HeLaN-ΔSLC cells but not in HeLaN-ΔB8 cells. This indicates that sulfated GAGs are the major attachment molecules on the surface of HeLa-N cells, while integrin αVβ8 contributes only minimally to the amount of virus attached under normal expression conditions. To further assess integrin-mediated binding in the absence of sulfated GAGs, we compared virus binding in HeLaN-ΔSLC, HeLaN-ΔSLCΔB8, and HeLaN-ΔSLC + human AVB8 cells. Although HeLaN-ΔSLC cells express endogenous integrin αVβ8 (Fig. 6c), the amount of virus attached was similarly low in both HeLaN-ΔSLC and HeLaN-ΔSLCΔB8 cells (Fig. 6b). This suggests that endogenous integrin αVβ8 provides only a very limited number of attachment sites, which is insufficient for detectable levels of virus attachment. In contrast, HeLaN-ΔSLC + human AVB8 cells showed a significant increase in virus binding (Fig. 6b), consistent with their elevated expression of integrin αVβ8 (Fig. 6c). This indicates that integrin αVβ8 can mediate SAFV-3 attachment on the cell surface in a manner dependent on its expression levels.

To confirm the specificity of these bindings, we examined the effect of pretreating viruses with soluble heparin or recombinant soluble integrin αVβ8 on SAFV-3 binding. Pretreatment with soluble heparin almost completely inhibited the binding of SAFV-3 to WT HeLa-N cells (Fig. 6d). This result further supported the idea that most of the virus binds to HeLa-N cells via sulfated GAGs. Next, we assessed the effect of pretreatment with soluble integrin αVβ8 on SAFV-3 binding in HeLaN-ΔSLC + human AVB8 cells (Fig. 6e). Pretreatment with soluble integrin αVβ8, but not soluble integrin αVβ3, inhibited viral binding in a dose-dependent manner. These results indicated that both sulfated GAGs and integrin αVβ8 mediate SAFV-3 attachment to the cell surface.

We also assessed the susceptibility of the cells to SAFV-3 infection (Fig. 6f) and compared it with their virus attachment profiles. HeLaN-ΔSLC cells showed reduced susceptibility, consistent with their reduced

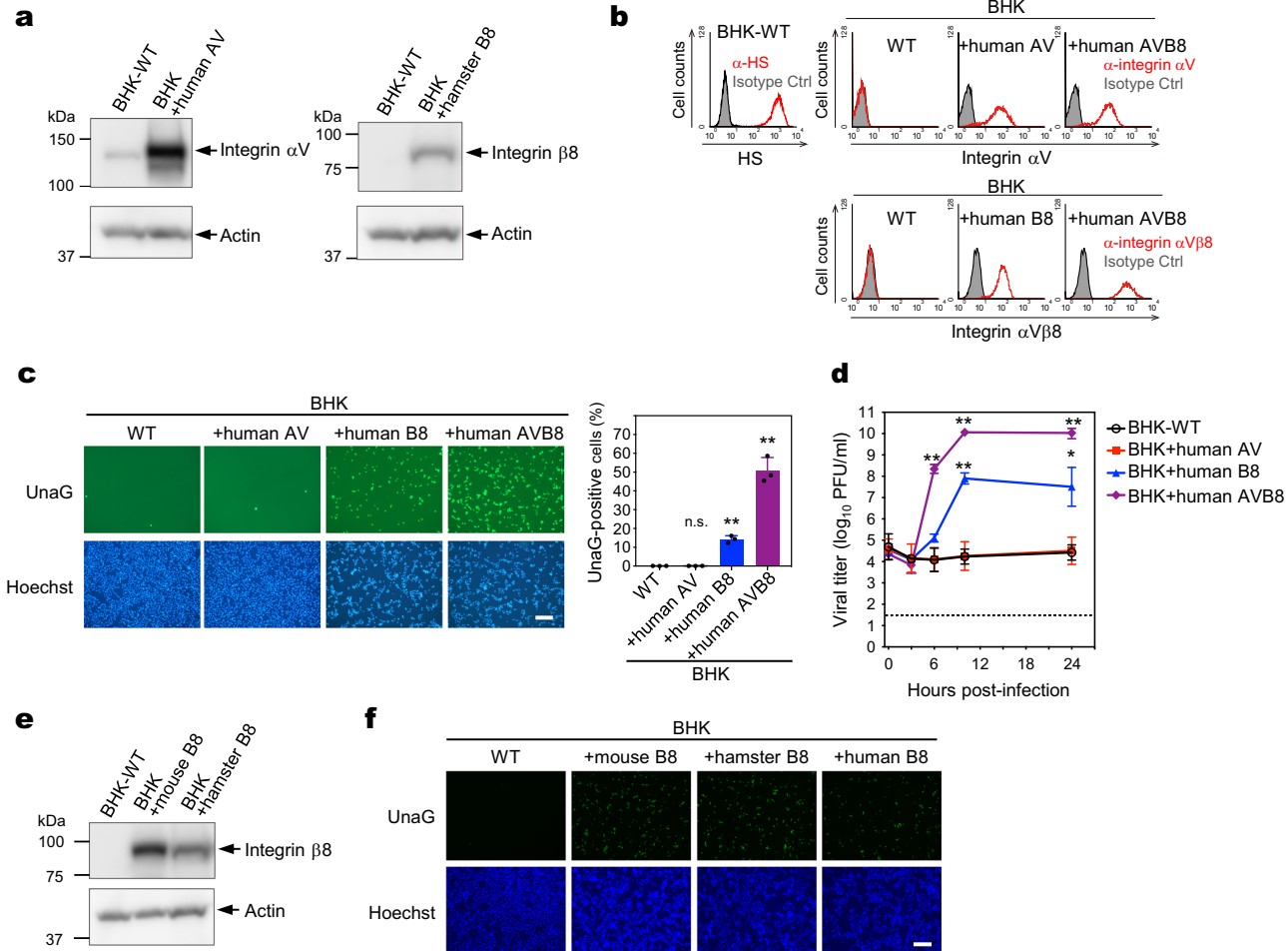

**Fig. 4 | SAFV-3 infection in BHK-21 cells expressing integrin β8 or αVβ8.**
**a** Western blot analysis of integrin αV (left panel) and integrin β8 (right panel) expression in BHK-21 cells. BHK-21 cells that were lentivirally transduced with either human integrin αV (BHK + human AV) or hamster integrin β8 (BHK + hamster B8) were used as positive controls. The anti-integrin αV antibody cross-reacted with both human and hamster integrin αV. Actin served as the loading control.
**b** Expression of HS, integrin αV, and β8 in BHK-21 derivatives. BHK-21 cells were stained with anti-HS antibody (upper left panel). BHK-21 cells stably expressing human integrin αV and/or β8 (BHK + human AV, BHK + human B8, BHK + human AVB8), as well as the control cells, were stained with anti-integrin αV or anti-integrin αVβ8 antibodies. The cells were analyzed by flow cytometry. **c** Susceptibility analysis using SAF/UnaG in BHK-21 cells expressing human integrin αV and/or β8. UnaG-positive cells (green, upper panel) and Hoechst-stained nuclei (blue, lower panel) were imaged at 16 h post-infection. Scale bar, 200 µm. The percentage of infected cells was determined by examining at least 1000 cells. Data are

representative of two independent experiments. **d** One-step growth kinetics of SAFV-3 in BHK + human AV, BHK + human B8, BHK + human AVB8, and control cells. The cells were infected with SAFV-3 and incubated for up to 24 h. The dotted line indicates the limit of detection. **e** Western blot analysis of exogenous integrin β8 expression in BHK-21 cells lentivirally transduced with either mouse or hamster integrin β8. The anti-integrin β8 antibody cross-reacted with both mouse and hamster integrin β8. Actin served as the loading control. **f** Susceptibility analysis using SAF/UnaG in mouse and hamster integrin β8 expressing BHK-21 cells. UnaG-positive cells (green, upper panel) and Hoechst-stained nuclei (blue, lower panel) were captured at 16 h post-infection. Scale bar, 200 µm. Images are representative of two independent experiments. Data in (**c** and **d**) represent means with s.d. ($n = 3$). Statistical significance was determined using a one-way ANOVA with Dunnett's multiple comparison test (**c**) and the two-sided Welch's t-test (**d**). **, $P < 0.01$, *, $P < 0.05$, n.s. not significant. Source data are provided as a Source Data file.

level of virus attachment, whereas HeLaN-ΔB8 cells exhibited a similar reduction in susceptibility despite having virus attachment levels comparable to those of WT cells. Notably, both HeLaN-ΔSLC and HeLaN-ΔB8 cells exhibited only about one-thousandth the susceptibility of WT cells. HeLaN-ΔSLCΔB8 cells were completely resistant, despite showing no further reduction in virus attachment compared to HeLaN-ΔSLC cells. HeLaN-ΔSLC + human AVB8 cells bound virus as effectively as WT cells but exhibited a 1000-fold higher susceptibility. These findings suggest that binding capacity and infection efficiency differ between the two entry routes: sulfated GAGs support high levels of virus attachment but mediate relatively inefficient productive infection, whereas integrin αVβ8 supports more efficient infection despite a lower binding capacity. The fact that infection still occurs in both HeLaN-ΔSLC and HeLaN-ΔB8 cells indicates that the sulfated GAGs and integrin pathways can function independently of each other. However,

the high susceptibility of WT cells cannot be fully explained by the sum of the two pathways alone, implying the existence of a cooperative mechanism in which sulfated GAGs act as primary attachment receptors that facilitate subsequent integrin-dependent entry.

## Sulfated GAGs enhance integrin αVβ8-mediated SAFV-3 infection

To confirm the cooperative function of sulfated GAGs and integrin αVβ8 in SAFV-3 infection, we analyzed the role of sulfated GAGs in the integrin αVβ8−dependent infection pathway using BHK-21 cells. While BHK-21 cells express sulfated GAGs, viral infection occurs only in the presence of exogenous integrin β8 (Fig. 4b, c). These findings indicate that viral binding to sulfated GAGs alone is not sufficient for infection in BHK-21 cells, making them an ideal model for analyzing the role of sulfated GAGs in integrin αVβ8-mediated SAFV-3 infection. First, we

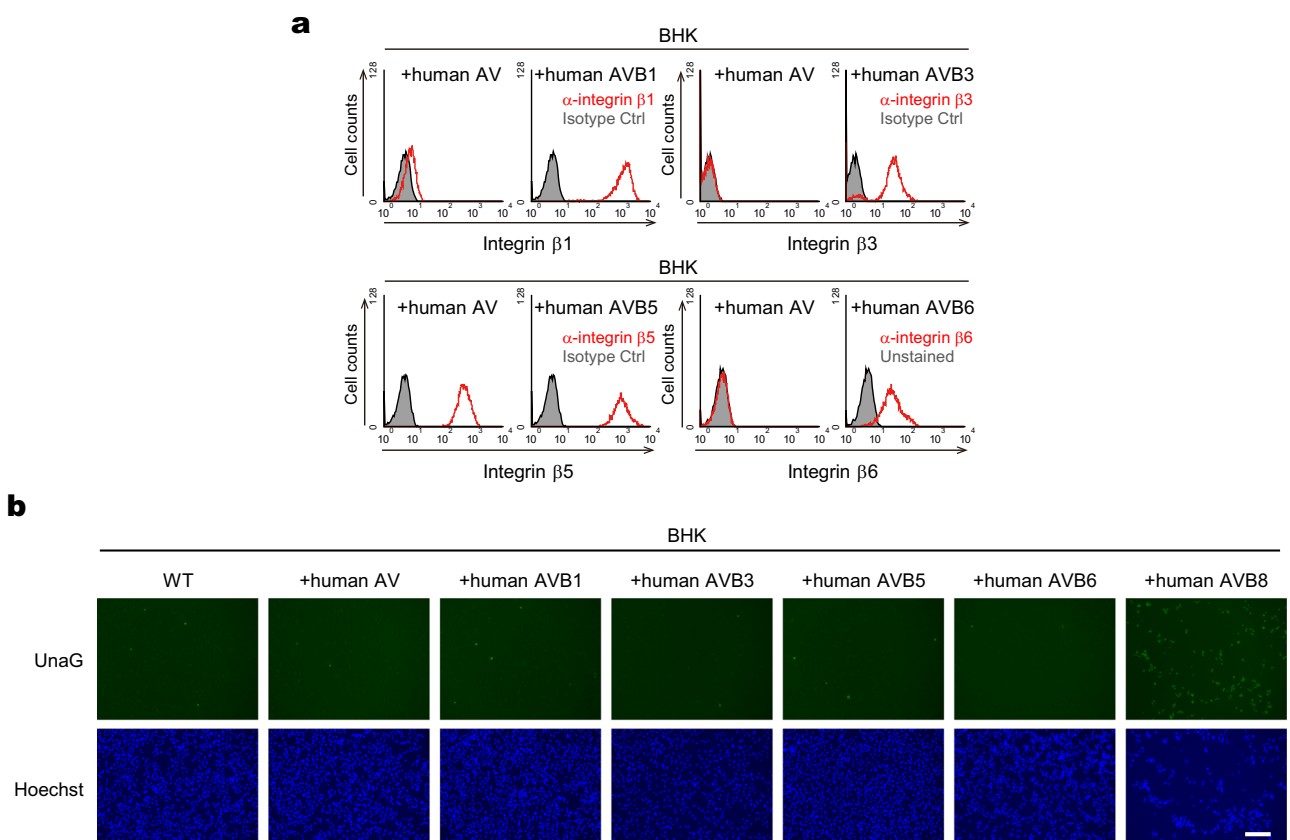

**Fig. 5 | SAFV-3 specifically utilizes integrin αVβ8 of the integrin αV subfamily for infection. a** Expression of human integrin β subunits (β1, β3, β5, and β6) on the surface of BHK-21 derivatives. BHK-21 cells lentivirally transduced with the respective integrin β subunits were stained with the indicated antibodies and analyzed using flow cytometry. **b** Susceptibility analysis using SAF/UnaG in human integrin αV and the indicated β subunit expressing BHK-21 cells. UnaG-positive cells (green, upper panel) and nuclei stained with Hoechst (blue, lower panel) were imaged at 16 h post-infection. Scale bar, 200 μm. Images are representative of two independent experiments.

established *SLC35B2* KO BHK-21 cells (BHK-ΔSLC) and confirmed the absence of HS on their surface as well as the introduction of frameshift mutations in the target region (Fig. 7a and Supplementary Fig. 7). SAFV-3 did not bind to BHK-ΔSLC cells (Fig. 7b). When human integrin αVβ8 (AVB8) was expressed in BHK-ΔSLC cells, where the integrin pathway is active, the cells acquired susceptibility to SAFV-3, albeit at lower levels than the BHK cells expressing human AVB8 (Fig. 7c). However, reintroduction of human *SLC35B2* partially restored HS expression (Fig. 7a) and significantly enhanced SAFV-3 susceptibility (Fig. 7c). These changes in susceptibility correlated well with the viral attachment mediated by sulfated GAGs (Fig. 7d). These results demonstrate that sulfated GAGs and integrin αVβ8 can function cooperatively, in addition to acting independently, during SAFV-3 infection.

This conclusion was further supported by observations in HeLa-R cells (RCB0007, RIKEN BRC), which we previously reported as being poorly susceptible to SAFV-3[19]. Flow cytometry revealed that HeLa-R cells express extremely low levels of HS compared with HeLa-N cells, whereas integrin αVβ8 expression is comparable (Supplementary Fig. 8). Thus, the reduced susceptibility of HeLa-R cells is likely due to insufficient HS expression, reinforcing the idea that efficient SAFV-3 infection requires adequate levels of both sulfated GAGs and integrin αVβ8.

### SAFV-3 binds to the RGD-binding site of integrin αVβ8

To determine whether the interaction between SAFV-3 and integrin αVβ8 is mediated by the RGD-binding site, we introduced mutations into this site and examined its effect. Latent-TGF-β, which contains an RGD motif, is known to bind to the RGD-binding site of integrin β8. A previous study reported that introducing specific mutations into this

site, without altering the overall three-dimensional structure of integrin β8, abolished its ability to bind latent-TGF-β[31]. Therefore, these mutations are predicted to also disrupt viral binding. To test this, we generated BHK-21 cells transiently expressing integrin β8 mutants: ΔSDL (a deletion mutant of "specificity-determining loop"); Y172N, and I208R (point mutants in RGD-binding site), which are known to lose their ability to bind latent-TGF-β[31], and inoculated them with SAFV-3. No viral propagation was observed in BHK-21 cells expressing the β8 mutants, whereas robust viral propagation was detected in WT integrin β8-expressing cells (Fig. 8a). These findings suggest that the SAFV-3-binding region of integrin αVβ8 is indeed an RGD-binding site.

Several picornaviruses use an RGD sequence in their capsids to bind integrins during infection[32–38]. Although SAFV-3 lacks a canonical RGD sequence, it carries an RAD (Arg-Ala-Asp) sequence in the puff A region of VP2, whereas SAFV-2 has an RLD (Arg-Leu-Asp) sequence in CD loop I of VP1 (Fig. 8b; detailed version in Supplementary Fig. 9). These regions, referred to as loops and puffs, protrude from the virion surface. However, RAD is generally thought to reduce integrin-binding affinity[39]. Since it is possible that the RAD sequence of SAFV-3 and the RLD sequence of SAFV-2 might still facilitate virus binding to integrin αVβ8, we investigated whether GRADS (Gly-Arg-Ala-Asp-Ser) and GRLDS (Gly-Arg-Leu-Asp-Ser) peptides could block SAFV infection by masking the RGD-binding site on integrin αVβ8. The GRGDS (Gly-Arg-Gly-Asp-Ser) peptide was used as a positive control and GRAES (Gly-Arg-Ala-Glu-Ser) was used as a negative control. Pretreatment of HeLaN-ΔSLC with the GRADS peptide reduced the number of UnaG-positive cells in a dose-dependent manner, suggesting that SAFV-3 may interact with integrin αVβ8 through the RAD sequence (Fig. 8c, left

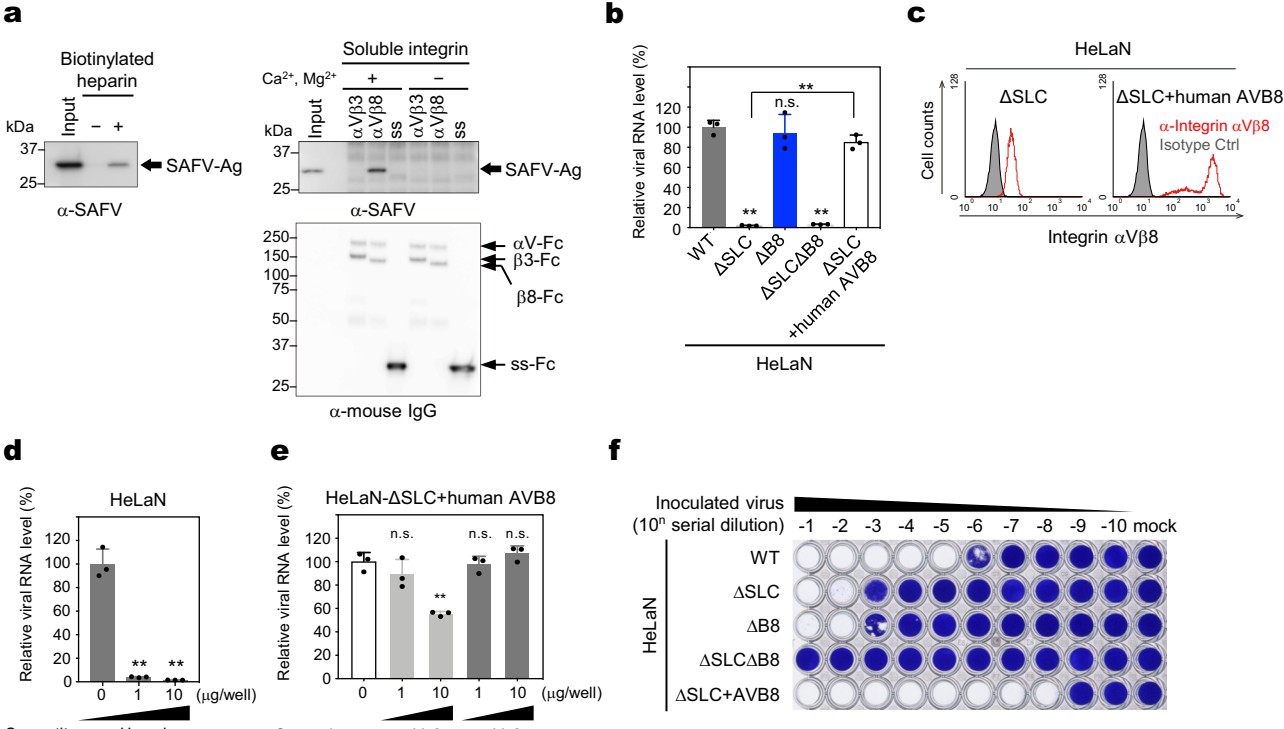

**Fig. 6 | SAFV-3 directly binds to HS and integrin αVβ8 on the cell surface. a** Pull-down assay of SAFV-3 using heparin (left panel) and integrin αVβ8 (right panel). Heparin and Fc chimera of extracellular domains of integrin αVβ8, αVβ3, or the signal sequence (ss) of integrin αV (negative control) were prepared as complexes with magnetic beads. These complexes were incubated with SAFV-3, followed by western blot analysis of the bound virus using anti-SAFV-3 antiserum (left and right upper panels). The bottom right panel shows an image of the integrin-Fc complex on magnetic beads used for pulldown, detected using an anti-mouse IgG antibody. **b** Cell surface attachment assay for SAFV-3. HeLaN-WT, HeLaN-ΔSLC, HeLaN-ΔB8, HeLaN-ΔSLCΔB8, and HeLaN-ΔSLC + human AVB8 were incubated with SAFV-3, followed by RT-qPCR analysis of the bound virus. **c** Expression of human integrin β8 in HeLaN-ΔSLC and HeLaN-ΔSLC + human AVB8 cells. To compare the expression levels of integrin αVβ8 on the cell surface, HeLaN-ΔSLC and HeLaN-ΔSLC + human AVB8 cells were stained with anti-integrin αVβ8 antibodies and analyzed by flow cytometry. **d, e** Inhibition of SAFV-3 attachment to the cell surface by soluble

heparin (**d**) or recombinant integrin αVβ8 (**e**). HeLaN-WT cells (**d**) or HeLaN-ΔSLC + human AVB8 cells (**e**) were incubated with SAFV-3 pretreated with 1 or 10 μg of soluble heparin or recombinant integrin αVβ8, respectively. Recombinant integrin αVβ3 was the negative control. After incubation at 4 °C for 2 h, bound virus was analyzed using RT-qPCR. **f** HeLaN-WT, HeLaN-ΔSLC, HeLaN-ΔB8, HeLaN-ΔSLCΔB8, and HeLaN-ΔSLC + human AVB8 cells were infected with tenfold serial dilutions of SAFV-3, and viable cells were stained with crystal violet to assess infection levels. All data are representative of two independent experiments. Data in (**b**, **d**, and **e**) represent means with s.d. ($n = 3$). Statistical significance was determined using a one-way ANOVA with Dunnett's multiple comparison test. **, $P < 0.01$, n.s. not significant. Asterisks directly placed on bars indicate statistically significant differences compared to WT or untreated samples, while asterisks placed on the lines connecting bars denote statistically significant differences between those bars. Source data are provided as a Source Data file. Ag antigen, Fc fragment crystallizable region.

panel). In contrast, pretreatment of HeLaN-ΔSLC with GRLDS peptide did not reduce the number of UnaG-positive cells (Fig. 8c, right panel), suggesting that SAFV-2 does not use the RLD motif to interact with integrin αVβ8. Since these results were inconclusive, we analyzed the mutant viruses. We introduced mutations into SAFV-3 and SAFV-2 to disrupt their respective RGD-like motifs (Fig. 8d) and examined the changes in infectivity. To exclude the effect of viral binding to sulfated GAGs, we used HeLaN-ΔSLCΔB8, HeLaN-ΔSLC, and HeLaN-ΔSLC + AVB8 cells. Mutations in the RGD-like motif did not affect the infectivity of either SAFV-3 or SAFV-2 (Fig. 8e).

These results indicated that SAFV-3 and SAFV-2 bind to the RGD-binding site of integrin αVβ8, but the viral integrin-binding site is not an RGD-like sequence.

## Discussion

Virus-host interactions that are critical for SAFV pathogenicity remain largely unknown. In the present study, our data demonstrated that double KO of *ITGAV* or *ITGB8* together with *SLC35B2* resulted in a complete loss of susceptibility to SAFV-3 and SAFV-2. In contrast, KO of either *SLC35B2* or integrin alone reduced susceptibility incompletely. Furthermore, we identified an interconnected pathway in which sulfated GAGs and integrin αVβ8 function together to facilitate infection.

Based on these findings, we proposed a model in which SAFV infection is mediated by dual and cooperative receptors (Fig. 9). However, how these pathways are utilized—whether simultaneously or individually—remains unclear. Notably, we observed strain-specific differences in the dependency on these pathways, which may contribute to variations in pathogenicity among the SAFV strains. In addition, the cellular context, such as the expression levels of sulfated GAGs and integrin αVβ8, may further influence which pathway is preferentially utilized, as suggested by the difference between HeLa-N and HeLa-R cells.

We employed a genome-wide CRISPR-Cas9 KO screen to identify SAFV receptors. However, the number of host factors identified in this screen was notably smaller than those reported for other viruses[13–15,17,18]. This is likely because, in the initial screening using SAFV-3 and HeLa-N cells, multiple infection pathways exist, making it rare for the knockout of a single gene to confer complete resistance. Furthermore, in the first screening in Fig. 1a, which was conducted under the strong selection pressure applied, only cells exhibiting relatively strong resistance to SAFV were selected, resulting in a limited set of candidate genes. In contrast, the second screening in Fig. 1a, conducted under milder selection pressure, yielded a larger population of surviving cells and, consequently, more candidate genes (see Source data for Fig. 1a). These observations suggest that the stringency of

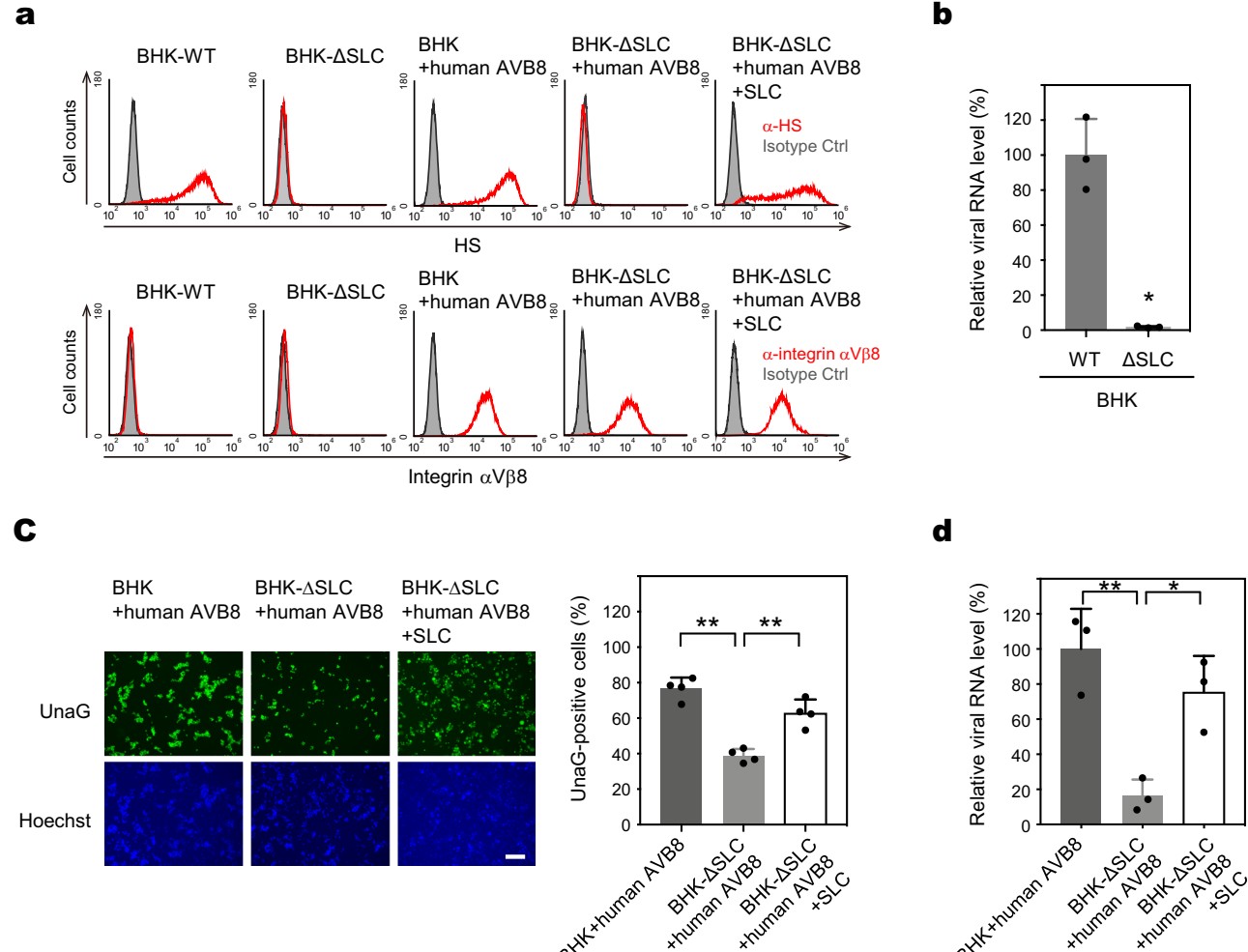

**Fig. 7 | Sulfated GAG enhances integrin αVβ8-mediated SAFV-3 infection.**
**a** Expression of HS and human integrin β8 in BHK-WT, BHK-ΔSLC, BHK + human AVB8, BHK-ΔSLC + human AVB8, and revertant cells expressing human *SLC35B2* (BHK-ΔSLC + human AVB8 + SLC). The cells were stained with anti-HS or anti-integrin αVβ8 antibodies and analyzed by flow cytometry. **b** Cell surface attachment assay for SAFV-3. BHK-WT and BHK-ΔSLC cells were incubated with SAFV-3 at 4 °C for 2 h. Viral binding to HS was assessed by RT-qPCR quantification of cell-bound virus (*n* = 3). **c** Susceptibility analysis using SAF/UnaG in BHK + human AVB8, BHK-ΔSLC + human AVB8, and BHK-ΔSLC + human AVB8 + SLC cells. The cells used in this experiment were sorted to equalize the surface expression levels of integrin

αVβ8 between BHK + human AVB8 and BHK-ΔSLC + human AVB8 cells. UnaG-positive cells (green) and nuclei stained with Hoechst (blue) were imaged at 16 h post-infection. The percentage of infected cells was determined by examining at least 800 cells/well (*n* = 4). Scale bar, 200 μm. **d** Cell surface attachment assay for SAFV-3 in BHK + human AVB8, BHK-ΔSLC + human AVB8, and BHK-ΔSLC + human AVB8 + SLC cells (*n* = 3). Bar graphs in (**b**–**d**) are presented as means with s.d. Statistical significance was determined using the two-sided Welch's *t*-test (**b**) and a one-way ANOVA with Dunnett's multiple comparison test (**c**, **d**). **, *P* < 0.01, *, *P* < 0.05. All data are representative of two independent experiments. Source data are provided as a Source Data file.

selection pressure applied during screening may influence the number of hits identified. Since the aim of this study was to identify genes essential for infection, we conducted two screenings under different selection stringencies and focused on genes consistently identified in both. In the secondary screening, which used two strains of SAFV-3 and *SLC35B2* KO cells, the low number of identified genes may be attributed to the presence of alternative molecules that support SAFV-3 replication. The identification of *PA2G4* as a common hit supports the notion that the screening functioned as intended. Our strategy in this study was to identify the genes most reliably enriched in the initial screening using HeLa-N cells, and then to identify additional essential genes in the secondary screening using SLC35B2 KO cells. Indeed, we successfully identified the integrins in the secondary screening, even though they were not consistently identified in the initial screening. We believe that this two-stage screening strategy was ultimately the most appropriate approach for this study.

Integrins function as heterodimers formed by combinations of α and β chains, with 18 known types of α chains, 8 types of β chains, and

24 confirmed αβ heterodimers on cell membranes[40]. Integrin αV is the most versatile subunit among α chains, forming heterodimers with five types of β subunits (β1, β3, β5, β6, or β8). Our study demonstrated that SAFV-3 specifically utilizes integrin αVβ8 for infection, distinguishing it from many other viruses that can interact with multiple β subunits. For example, parechovirus A1 (PeV-A1) uses αVβ1, αVβ3, and αVβ6[32–34], coxsackievirus A9 (CVA9) uses αVβ3 and αVβ6[35,36], and FMDV uses αVβ1, αVβ3, αVβ6, αVβ8, and α5β1 as receptors[37,41]. In addition, our results showed that both mouse and hamster integrin β8 function as receptors for SAFV-3 infection, similarly to human integrin β8 (Fig. 4f). This finding suggests that the β8 subunits of human and rodent integrins are interchangeable and that species-specific differences in integrin β8 do not affect SAFV infection.

Integrin heterodimers containing the αV subunit (αVβ1, αVβ3, αVβ5, αVβ6, αVβ8) are known as "RGD receptors" due to their ability to bind to proteins with an RGD motif, such as vitronectin and fibronectin[42,43]. Recently, sialylated integrins (αXβ2 and αMβ2) were identified as receptors for TMEV-DA strain infection[38]. FMDV utilizes

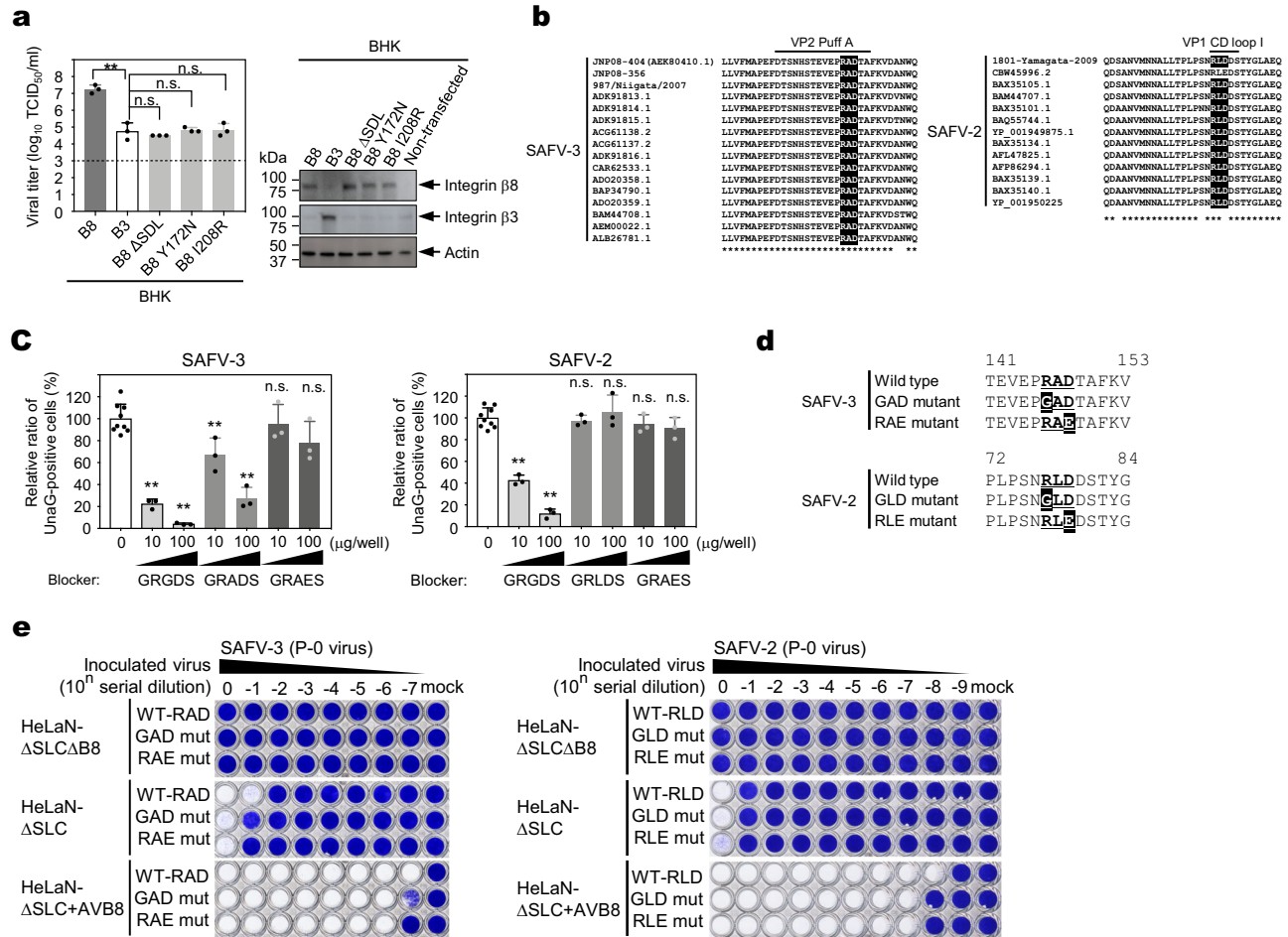

**Fig. 8 | SAFV binds to the RGD-binding site of integrin αVβ8 via an alternative sequence rather than an RGD-like sequence. a** Viral infection analysis using integrin β8 mutants. BHK-21 cells expressing the human integrin β8 mutants (ΔSDL, Y172N, and I208R) were inoculated with SAFV-3. After 2 days, virus titers were determined using the TCID50 assay. The dotted line indicates the limit of detection. Human integrin β3 was the negative control. The right panel shows the results of western blot analysis of integrin β8 mutants and integrin β3 expression in BHK-21 cells. **b** Alignment of the amino acid sequences of puff A on VP2 of SAFV-3 (left panel) and CD loop I on VP1 of SAFV-2 (right panel). RGD-like sequences are highlighted. **c** Infection blocking assay using RGD peptide. Left panel: HeLaN-ΔSLC cells were pretreated with 10 or 100 μg of GRGDS, GRADS, or GRAES peptide at 4 °C for 30 min and then incubated with SAFV-3/UnaG virus for an additional 60 min. Right panel: HeLaN-ΔSLC cells were pretreated with 10 or 100 μg of GRGDS, GRLDS, or GRAES peptide for 30 min and then incubated with SAFV-2/UnaG virus for an

additional 60 min. GRGDS and GRAES peptides were positive and negative controls, respectively. The number of UnaG-positive cells at 14 h post-infection was counted using ImageJ software. **d** Schematic illustration of mutagenesis in RGD-like sequence. The mutated amino acid residues are highlighted. **e** HeLaN-ΔSLCΔB8, HeLaN-ΔSLC, and HeLaN-ΔSLC + human AVB8 cells were infected with tenfold serial dilutions of mutant viruses carrying mutations in the RGD-like sequences, and viable cells were stained with crystal violet. Primary progeny virus produced from BHK cells transfected with infectious RNA (P-0 virus) was used. Bar graphs in (**a** and **c**) represent means with s.d. (*n* = 3 in (**a**); *n* = 9 peptide-free and *n* = 3 peptide-added samples in (**c**)). Statistical significance was determined using a one-way ANOVA with Dunnett's multiple comparison test. **, *P* < 0.01, n.s. not significant. Statistical comparisons in (**c**) were made between peptide-free and peptide-added samples. All data are representative of two independent experiments. Source data are provided as a Source Data file.

the five types of integrin heterodimers described above, and for both TMEV-DA and FMDV, the RGD motif on the protruding loop of the capsid protein plays a key role in integrin binding[37,38]. Similar RGD motif-dependent interactions between virions and integrins have also been observed in PeVA1 and CVA9[32–36]. However, the capsid proteins of SAFV-3 and SAFV-2 lack the RGD motif. Our data from the integrin β8 mutant analysis (Fig. 8a) and GRGDS peptide-blocking assay (Fig. 8c) clearly demonstrated that the virus binding region of integrin αVβ8 is the RGD-binding site. In contrast, the GRADS peptide blocked SAFV-3 infection, whereas viruses carrying mutations in the RAD sequence did not show reduced infectivity. This was presumably because the alternative sequence for binding to integrins in SAFV-3 had a lower affinity for integrins than the GRADS peptide. In contrast, SAFV-2 infection was not inhibited by the GRLDS peptide, nor was the infectivity of viruses with mutations in the RLD sequence significantly reduced. These results indicated that the RGD-like sequences of SAFV-3 and SAFV-2

were not integrin-binding sites. These findings suggest that, while SAFV binds to the RGD-binding site of integrin αVβ8, this interaction does not involve RGD-like sequences on the SAFV capsid (RAD in SAFV-3 and RLD in SAFV-2). Instead, an alternative sequence is responsible for integrin binding. Identifying this specific binding site in the virion requires further structural analyses, such as X-ray crystallography or cryo-electron microscopy.

SLC35B2 not only plays a role in heparan sulfate sulfation but also regulates overall protein tyrosine sulfation[16,23–25]. When we compared SAFV-3 susceptibility in *EXT1* KO and *SLC35B2* KO cells, no significant difference was observed, indicating that the reduced SAFV-3 susceptibility in *SLC35B2* KO cells was primarily due to the loss of sulfated GAGs, with overall protein tyrosine sulfation playing a negligible role. Cell surface sulfated GAGs, including HS, are ubiquitous negatively charged molecules that are commonly used by many viruses as attachment and entry receptors[44–46]. In HeLa-N cells, KO of *SLC35B2*

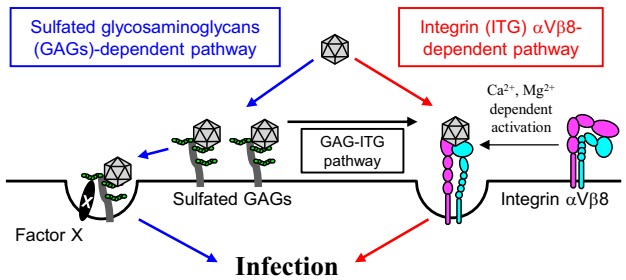

**Fig. 9 | A model for SAFV infection.** Sulfated GAGs and integrin αVβ8 function as interconnected dual receptors for SAFV infection in HeLa-N cells. SAFV can directly bind to either sulfated GAGs or integrin αVβ8, while a portion of viruses bound to sulfated GAGs subsequently interact with integrin αVβ8. In addition, the data suggest the existence of a downstream molecule (factor X) required for an unspecified step in the viral entry process following sulfated GAGs binding.

resulted in reduced susceptibility to SAFV-3 infection (Fig. 1c, d). BHK-21 cells, which express HS on their surface, were not susceptible to SAFV-3 (Fig. 4c, d), indicating that binding to HS alone was insufficient for SAFV infection. Given that SAFV-3 infection still occurred through sulfated GAGs even in integrin-deficient HeLa-N cells (Fig. 2c, d), this suggests the presence of an additional factor X involved in the post-adsorption process in the sulfated GAG-mediated pathway of SAFV infection in HeLa-N cells, but not in BHK-21 cells. Identifying this factor, which operates downstream of the interaction between SAFV and sulfated GAGs but is independent of integrin αVβ8, remains an important future research focus (Fig. 9). Conversely, certain picornaviruses develop an increased affinity for HS when passaged in cultured cells[44,47–51]. Our infectivity experiments with various clinical isolates and genotypes of SAFV revealed differences in receptor dependency (sulfated GAGs vs. integrin) among the strains (Fig. 3). However, the correlation between sulfated GAG dependency and adaptation to cultured cells remains unclear, as virus isolation methods vary across different institutions (Supplementary Table 1). Given that receptor dependency differs by strain (Fig. 3), it is possible that sulfated GAG-mediated infection resulted from adaptation to culture conditions, similar to other picornaviruses. To clarify this, future studies should examine receptor usage in viruses that have not adapted to the cultured cells. In addition, because this study was limited to in vitro cell line models, the relevance of these findings to in vivo infection remains uncertain. Future analyses using primary human cell models are crucial for gaining further insights into the pathogenesis of SAFV.

In summary, this study identified integrin αVβ8 and sulfated GAGs as key receptors for SAFV infection and demonstrated their independent and cooperative functions. Our findings suggest that sulfated GAGs primarily serve as attachment receptors, whereas other factors play a role in the subsequent process. Although integrin αVβ8 is clearly involved in attachment, its role in the later stages of infection, such as entry and uncoating, remains unclear. Further elucidation of these mechanisms is essential for a deeper understanding of the SAFV pathogenesis.

## Methods
### Cells
HeLa-N is a HeLa subline that is highly susceptible to SAFV-3, whereas HeLa-R is a less susceptible subline obtained from RIKEN BRC (RCB0007)[19]. HeLa-N and 293 T[52] cells were maintained in Dulbecco's Modified Eagle's Medium (DMEM; Nacalai Tesque) supplemented with 10% fetal calf serum (FCS) containing 100 U/ml penicillin and 100 μg/ml streptomycin. HeLa-R cells were maintained in Eagle's Minimum Essential Medium (MEM; Nissui) supplemented with 10% calf serum (CS) and 0.03% ʟ-glutamine. BHK-21 (C-13) cells were obtained from the JCRB Cell Bank (JCRB9020). BHK-21[53] and BHK-21 (C-13) cells

were maintained in MEM supplemented with 5% CS and 0.03% ʟ-glutamine. Caco-2 cells[54] were maintained in MEM supplemented with 20% FCS, 0.03% ʟ-glutamine, and 0.1 mM non-essential amino acids (Gibco). RD-18S-Niigata (RD-18S-N)[55] cells were maintained in MEM with 8% FCS, 0.03% ʟ-glutamine, and 0.5% Fangizon. All the cell lines were incubated at 37 °C in a humidified atmosphere containing 5% $CO_2$.

### Viruses
SAFV genotype 3 (SAFV-3) was prepared from the infectious cDNA clone pSAF404 derived from the JPN08-404 strain[7]. The virus was propagated in HeLa-N cells, and the viral RNA sequence was confirmed to be identical to that of the pSAF404 cDNA clone. The SAF/UnaG virus was prepared as previously reported[26]. The other SAFV-3 strains used in this study were as follows: JPN08-356, isolated from pharyngeal swabs and kindly provided by Dr. Hiroyuki Shimizu at the National Institute of Infectious Diseases; and 987/Niigata/2007[6], isolated from cerebrospinal fluid and kindly provided by Dr. Chika Hirokawa at the Niigata Prefectural Institute of Public Health and Environmental Sciences. These viruses were also propagated in HeLa-N cells. The SAFV-2 isolate (1801-Yamagata-2009), isolated from pharyngeal swabs, was kindly provided by Dr. Katsumi Mizuta at the Yamagata Prefectural Institute of Public Health and propagated in RD-18S-N cells[55,56]. Viral titers were determined using a standard plaque assay in HeLa-N cells. An infectious cDNA clone of SAFV-2 was constructed from the 1801-Yamagata-2009 strain using the same method as that for pSAF404[7]. Infectious cDNA clones of mutant viruses carrying mutations in the RAD sequence of SAFV-3 and RLD sequence of SAFV-2 were prepared by overlap PCR using the respective infectious cDNA clones as backbones. Infectious RNAs synthesized by in vitro transcription were transfected into BHK-21 cells, and the viruses produced were harvested 18 h post-transfection. In the case of SAFV, progeny viruses produced from RNA-transfected BHK-21 cells do not infect neighboring BHK-21 cells, allowing the collection of viruses that have not been subjected to selection pressure from the receptor. EMCV was provided by Dr. Takashi Fujita (Kyoto University). CVB3 (Nancy strain) was derived from an infectious cDNA clone provided by Dr. Reinhardt Kandolf[57].

### Genome-wide CRISPR-Cas9 knockout screen
We screened the genes required for SAFV-3 infection using a previously described method with some modifications[20]. Briefly, for genome-wide CRISPR-Cas9 knockout (KO) screening, we used the GeCKO v2.0 two-vector system (Addgene #1000000049). HeLa-N and HeLaN-ΔSLC cells were stably transduced with lentiCas9-Blast (Addgene #52962) and subsequently selected in medium containing 4 μg/ml Blasticidin S for 7 days. Next, 210 million HeLa-N and HeLaN-ΔSLC cells constitutively expressing Cas9 were transduced with lentiGuide-Puro from the human CRISPR KO pooled lentiviral library at a multiplicity of infection (MOI) of 0.3 in the presence of 8 μg/ml polybrene. The cells were then selected using 1 μg/ml puromycin for 7 days. For CRISPR screening, 30 million cells were infected with SAFV-3 (JPN08-404 and JPN08-356 strains) at a high MOI, causing 95% cytopathic effects (CPE) in untransduced HeLa-N cells within 24 h and in untransduced HeLaN-ΔSLC cells within 48 h. Surviving cells were subjected to a second infection at a high MOI. During the initial screening with HeLa-N cells, we adjusted the selection pressure. Under strong selection conditions, viruses were retained in the culture medium for 25 days until the cells had proliferated sufficiently for subsequent analysis. During this period, the medium was replaced every four days to maintain the concentration of the selective drug, but the cells were not washed to avoid complete removal of the virus. Under mild selection conditions, viruses were removed 24 h post-infection upon observation of CPE, and to minimize excessive selection pressure and preserve cell viability, the cells were washed with PBS (−) and the culture medium was replaced daily. Genomic DNA was

extracted from the pooled surviving cells using the Blood & Cell Culture DNA Maxi Kit (13362, Qiagen) or Puregene Cell Kit (158043, Qiagen). sgRNA sequences were amplified from the genomic DNA using NEBNext High-Fidelity 2× PCR Master Mix (M0541S, NEB) with 10 NGS-Lib-Fwd primers and one barcoded NGS-Lib-KO-Rev primer[20]. The PCR product was concentrated by isopropanol precipitation and electrophoresed on 3% agarose gel. DNA between 200 and 300 base pairs was extracted using a QIAquick Gel Extraction Kit (28704, Qiagen) and further purified using AMPure XP (A63880, Beckman Coulter). Sequencing of the PCR products was performed using next-generation sequencing (NGS; Illumina MiSeq instrument; Illumina). The acquired FASTQ files were cleaned by trimming and removing low-quality reads using the PRINSEQ software program, version 0.20.4 (http://prinseq.sourceforge.net/index.html). Count data for sgRNA were extracted from FASTQ files using count_spacers.py[20]. We searched for genes for which the number of remaining sgRNAs was significantly different between uninfected samples and those infected with SAFV-3, using MAGeCK version 0.5.9.5[58].

### Lentiviral transduction

For lentivirus production, 293 T cells were transfected with the lentiviral plasmid and packaging plasmids (pCAG-HIVgp and pCMV-VSV-G-RSV-Rev from RIKEN BRC) using PEI Max transfection reagent (24765, Polysciences) and cultured for 72 h. The resulting lentivirus-containing culture supernatant was added to the target cells with 8 µg/ml polybrene. The cells were then cultured in selection medium supplemented with the appropriate antibiotics: 1 µg/ml puromycin (ant-pr-1, InvivoGen), 1 mg/ml G418 (074-05963, Fujifilm Wako Pure Chemical), and 10 µg/ml Blasticidin S (ant-bl-1, InvivoGen) for HeLa-N and 20 µg/ml Blasticidin S, 800 µg/ml G418, and 400 µg/ml Zeocin (R25001, Thermo Fisher Scientific) for BHK-21 cells.

### Establishment of KO cell lines

To establish the *SLC35B2* KO cell lines (HeLaN-ΔSLC and BHK-ΔSLC), the pGuide-it-ZsGreen1 plasmid (632601, Takara Bio) containing either the human (5′- AGAGTGATGACCCGCAGCTA -3′) or hamster (5′-GCTCGCCGCGCTCCCGTCTT -3′) *SLC35B2* sgRNA sequences was transfected into HeLa-N and BHK-21 cells, respectively, using Lipofectamine 2000 transfection reagent (11668019, Thermo Fisher Scientific). For the *EXT1* KO HeLa cell line (HeLaN-ΔEXT1), the pGuide-it-ZsGreen1 plasmid containing the *EXT1* sgRNA sequence (5′- CGCAGAGCGTCCGGGAAGCG -3′) was used. Forty-eight hours after transfection, ZsGreen-positive cells were sorted using an SH800 cell sorter (SONY) and cloned using limiting dilution. ΔSLC and ΔEXT1 clonal cells were assessed for HS expression on the cell surface using flow cytometry (see below). The genomic DNA surrounding the target sequence was amplified by PCR and sequenced by Sanger sequencing. Chromatograms were analyzed using DECODR v3.0[59] (https://decodr.org/) and TIDE[60] (https://tide.nki.nl/).

To establish *ITGAV* and *ITGB8* KO cell lines (HeLaN-ΔAV, HeLaN-ΔB8, HeLaN-ΔSLCΔAV, and HeLaN-ΔSLCΔB8), sgRNA sequences (*ITGAV*: 5′- AAATTCCAATGGATCATCCT -3′, *ITGB8*: 5′- CAAATGCAGCATCCTGTGCC -3′) were cloned into lentiGuide-Puro (Addgene #52963). The lentivirus was added to Cas9-expressing HeLa-N and HeLaN-ΔSLC cells, which were cultured in selection medium containing 1 µg/ml puromycin for 7 days and then cloned by limiting dilution. To confirm the depletion of integrins, cell surface expression was examined by flow cytometry (see below). Genomic DNA surrounding the target sequence was PCR-amplified and analyzed as described above.

### Flow cytometry

Cells were detached using Accutase (12679-54, Nacalai Tesque) and incubated with primary antibodies for 30 min on ice, followed by incubation with secondary antibodies for an additional 30 min on ice if

necessary. The cells were analyzed using a FACS Canto II (BD Biosciences) or a DxFLEX (Beckman Coulter) flow cytometer, and data were processed using WinMDI or Kaluza (Beckman Coulter) software. To detect HS, the cells were stained with biotinylated mouse anti-heparan sulfate antibody (10E4 epitope) (370255-B, amsbio) or biotinylated mouse IgMκ isotype control (401621, BioLegend), followed by PE-conjugated streptavidin (405203, BioLegend) staining. For the detection of human integrin αV, β1, β3, β5, and β6, cells were stained with PE-conjugated antibodies against CD51 (integrin αV) (327910, BioLegend), CD29 (integrin β1) (303003, BioLegend), CD61 (integrin β3) (336405, BioLegend), integrin β5 (345203, BioLegend), and an APC-conjugated antibody against integrin β6 (FAB4155A, R&D Systems). PE-conjugated mouse IgG1κ (981804, BioLegend) and IgG2aκ (400213, BioLegend) isotype-matched antibodies were used as controls. For human integrin β8 detection, the cells were stained with anti-integrin αVβ8 (clone EM13309) (ZRB1192, Sigma Aldrich), followed by PE-conjugated donkey anti-rabbit IgG secondary antibody (406421, BioLegend). Rabbit polyclonal isotype antibody (910801, BioLegend) was used as a control.

### Cloning

Total RNA was extracted from HeLa-N, Caco-2, and BHK-21 (C-13) cells using an RNeasy Mini Kit (74104, Qiagen) following the manufacturer's instructions. cDNAs for human *ITGAV* and *ITGB8* from HeLa-N cells, human *ITGB1* and *ITGB5* from Caco-2 cells, and hamster *Itgb8* from BHK-21 (C-13) cells were synthesized and amplified using the PrimeScript II High Fidelity One Step RT-PCR Kit (R026A, Takara Bio). cDNAs for human *ITGB3* and *ITGB6* and mouse *Itgb8* were PCR-amplified from the plasmids MHS6278-211691048, MHS6278-211690966, and MMM1013-211691535, respectively (Horizon). The cDNAs were subsequently cloned into the pENTR-2B or pENTR/D-TOPO vector (K240020SP, Thermo Fisher Scientific) and transferred to the lentiviral expression vectors CSII-EF-IN-RfA[18] or CSII-EF-IB-RfA[52] using LR Clonase (11791020, Thermo Fisher Scientific). *SLC35B2* cDNA was amplified from Caco-2 RNA and ligated into CSII-PGK-IZ, in which the human PGK promoter and IRES-zeocin resistance gene cassette were assembled into the backbone of the lentiviral vector plasmid CS-CDF-CG-PRE (RIKEN BRC). Fc-tagged integrins, consisting of the extracellular domains of integrin αV, β8, or β3 fused to the mouse IgG2a Fc tag from pFUSEss-CHIg-mG2a_M18 (105930, Addgene), were cloned into the pENTR-2B vector and subsequently transferred to the expression vector pEFneo-RfA[61] using LR Clonase, resulting in pEFneo-ITGαV-Fc, pEFneo-ITGβ8-Fc, and pEFneo-ITGβ3-Fc. pEFneo-ss-Fc, containing only the signal sequence (ss) of integrin αV, was constructed in the same manner. Full-length cDNA of human *ITGB8* and *ITGB3* with a FLAG tag at the 3′ end was cloned into the *Eco*RI site of pCAGGS-PUR (pCAG-hITGβ8-F and pCAG-hITGβ3-F)[62]. Mutations that disrupt the RGD-binding site (ΔSDL, Y172N, and I208R) are described in ref. 31. The same mutations were introduced into the pCAG-hITGβ8-F.

### Viral susceptibility analysis

To examine the susceptibility of KO cell lines to SAFV-3 (cDNA-derived JPN08-404) and SAFV clinical isolates, 50 µl of tenfold serial dilutions of viruses and 100 µl of $8 \times 10^3$ cells were added to each well of uncoated or collagen-coated 96-well plates and incubated at 37 °C with 5% $CO_2$ for 4 or 5 days. The cells were subsequently fixed with 10% neutral-buffered formalin and stained with a 1% crystal violet/20% methanol solution. When using the SAF/UnaG virus, $4 \times 10^5$ cells were seeded in collagen-coated 6-well plates and incubated at 37 °C for at least 24 h. The cells were then washed with phosphate-buffered saline (PBS) (−) and inoculated with 200 µl of SAF/UnaG ($8 \times 10^6$ PFU/well) for 1 h at 37 °C with 5% $CO_2$. After inoculation, the cells were washed with serum-free DMEM and incubated at 37 °C in 1 mL 1% FCS DMEM. At 16 h post-infection, UnaG expression in the infected cells was captured by fluorescence microscopy with nuclear staining using Hoechst 33342.

For BHK-21 and its derivatives, 50 μl of SAF/UnaG virus ($2 \times 10^5$ PFU/well, diluted in 5% CS MEM) and 100 μl of $2 \times 10^4$ cells were added to each well of 96-well plates and incubated at 37 °C. After 16 h of incubation, UnaG expression in the infected cells was captured using fluorescence microscopy, with nuclear staining using Hoechst 33342 when necessary. The number of UnaG-positive cells was quantified using ImageJ software. The PFU values of viral titers in the experiments using BHK-21 cells were determined in HeLa-N cells.

## Viral growth kinetics

To measure SAFV-3 growth in HeLa-N cells and their derivatives, cells were seeded at a density of $1 \times 10^5$ cells/well in 24-well plates and cultured for one day. The cells were then inoculated with SAFV-3 at $1 \times 10^4$ PFU/well and incubated at 37 °C with 5% $CO_2$ for 0, 1, 2, 3, 4, or 5 days before collection. Samples were prepared by subjecting the cells to three freeze–thaw cycles to release virions, followed by centrifugation to remove cell debris. Viral titers were determined in HeLa-N cells by the 50% tissue culture infectious dose ($TCID_{50}$) assay using the Kärber method[63]. BHK-21 cells and their derivatives were seeded at a density of $4 \times 10^5$ cells/well in 6-well plates and cultured for 1 day. After washing the cells with PBS (−), they were inoculated with 200 μl of SAFV-3 ($8 \times 10^6$ PFU/well) and incubated for 1 h at 37 °C with 5% $CO_2$. After inoculation, the cells were washed twice with PBS (−) and incubated at 37 °C in 1 mL of 1% CS MEM. The cells and supernatants were collected at 0, 3, 6, 10, and 24 h post-infection. Samples were prepared by three freeze–thaw cycles and centrifuged to remove cell debris. Viral titers were determined using a standard plaque assay with HeLa-N cells. The PFU values were also used in experiments using BHK-21 cells, as described above.

## Western blotting

Cells were lysed using 1× sodium dodecyl sulfate-polyacrylamide gel electrophoresis (SDS−PAGE) sample loading buffer (786-701, G-Biosciences) supplemented with 2-mercaptoethanol and complete mini protease inhibitor cocktail tablets (11836153001, Roche) and boiled for 5 min. Samples were separated by SDS−PAGE using 10−20% Extra PAGE One precast gels (13068-24, Nacalai Tesque) and transferred onto a polyvinylidene difluoride (PVDF) membrane (Trans-Blot Turbo Mini PVDF Transfer Packs; 1704156, Bio-Rad Laboratories) using the Trans-Blot Turbo Transfer System (1704150, Bio-Rad Laboratories). The membrane was blocked with 5% skim milk in TBS-T (T9142, Takara Bio) and incubated with primary and secondary antibodies. To detect of endogenous hamster integrin αV and β8, rabbit anti-human integrin αV polyclonal antibody (27096-1-AP, Proteintech) and rabbit anti-mouse integrin β8 (D1V7M) monoclonal antibody (88300, Cell Signaling Technology) were used, respectively. To detect the expression of human integrin β8 and β3, rabbit anti-integrin αVβ8 monoclonal antibody (clone EM13309) and rabbit anti-integrin αVβ3 monoclonal antibody (clone EM22703) (ZRB1190, Sigma-Aldrich), respectively, were used. As a loading control, actin was detected using an anti-actin (AC-40) monoclonal antibody (A3853, Sigma Aldrich). SAFV antigen (SAFV-Ag) was detected using rabbit anti-SAFV-3 antiserum[19]. The secondary antibodies used were horseradish peroxidase-conjugated anti-mouse and anti-rabbit IgG (170-6516 and 170-6515, respectively; Bio-Rad Laboratories). Signals were detected using ECL Prime Western Blotting Detection Reagent (RPN2236, Cytiva).

## Viral attachment and inhibition assay

HeLa-N cells and their derivatives were seeded at $5 \times 10^4$ cells/well in 24-well plates and cultured for 2 days. The cells were then incubated with SAFV-3 ($2 \times 10^6$ PFU/well) at 4 °C for 2 h to allow viral attachment. BHK−21 cells and their derivatives were seeded at a density of $1.5 \times 10^5$ cells/well in 6-well plates and cultured for 2 days. The cells were incubated with SAFV-3 ($1.6 \times 10^7$ PFU/well) at 4 °C for 2 h to allow viral

attachment. Following adsorption, the cells were washed three times with cold medium containing serum to remove unbound viruses. Total RNA was extracted using the RNeasy Mini Kit and bound viral RNA was quantified by RT-qPCR. For the viral binding inhibition assay, SAFV-3 ($1 \times 10^6$ PFU/well) was pretreated with either 1 μg or 10 μg of heparin sodium salt (17513-96, Nacalai Tesque) or recombinant human integrin αVβ8 (4135-AV, R&D Systems). Recombinant human integrin αVβ3 (3050-AV, R&D Systems) was used as a negative control. The mixtures were incubated at 4 °C for 2 h and then inoculated into cells that had been pre-seeded in 24-well plates. After adsorption at 4 °C for 2 h, the cells were washed three times with cold DMEM containing 10% FCS. The bound viral RNA was analyzed by RT-qPCR using either the One Step TB Green PrimeScript RT-PCR Kit II (RR086A, Takara Bio) with a QuantStudio 12 K Flex Real-Time PCR System (Applied Biosystems), or PrimeScript RT Master Mix (RR036A, Takara Bio) and the THUNDER-BIRD SYBR qPCR Mix (QPS-201, TOYOBO) with a StepOnePlus Real-Time PCR Systems Applied Biosystems (Thermo Fisher Scientific). The following primers were used: SAFV forward primer, 5′-TGTAGC-GACCTCACAGTAGCAG-3′ and SAFV reverse primer, 5′-AGGA-CATTCTTGGCTTCTCTACCG-3′. For the blocking assay using RGD peptide, cells were seeded at $1.5 \times 10^4$ cells/well in 48-well plates and cultured for 2 days. The cells were pre-incubated with the synthetic peptides GRGDS (4189, Peptide Institute), GRAES, GRADS, or GRLDS (custom-synthesized, GL Biochem Ltd.) at 4 °C for 30 min. Next, 50 μl of SAF/UnaG (12,000 UnaG-positive units in HeLaN-ΔSLC cells) was added and incubated at 37 °C for 1 h (total volume of 300 μl). The cells were subsequently washed three times with DMEM containing 10% FCS and incubated in the growth medium at 37 °C for 14 h. UnaG expression in infected cells was observed by fluorescence microscopy with nuclear staining using Hoechst 33342. The number of UnaG-positive cells was quantified using ImageJ software.

## Integrin β8 mutant analysis

BHK-21 cells ($1 \times 10^5$ cells/well) were seeded into 24-well plates. The next day, plasmids expressing WT human integrin β8 (pCAG-hITGβ8-F) and its mutants (pCAG-hITGβ8-ΔSDL, pCAG-hITGβ8-Y172N, pCAG-hITGβ8-I208R) as well as pCAG-hITGβ3-F as a negative control, were transfected in triplicate (0.5 μg/well) using FuGENE HD (E2311, Promega). One day after transfection, SAFV ($1 \times 10^4$ PFU/well) was inoculated for 2 h, cells were washed once, and fresh medium was added. The cells were incubated at 37 °C. Two days later, the cells and supernatants were collected and frozen, and the viral titers were determined.

## Pull-down assay

To prepare heparin-biotin-streptavidin (SA) magnetic bead complexes, 50 μl of SA-magnetic beads (S1420, NEB) were incubated with 400 μg (25 μl) of biotinylated heparin (B9806, Sigma Aldrich) at 4 °C for 1 h with rotation. For the pull-down of SAFV-3 by heparin, SAFV-3 ($1 \times 10^6$ PFU) was mixed with 50 μl of heparin-biotin-SA beads or biotin-SA beads without heparin (negative control). The mixture was then incubated at 4 °C for 1 h with rotation. After incubation, the beads were washed three times with 0.1% BSA-PBS (−). The precipitates were boiled in 2× SDS−PAGE sample loading buffer and subjected to western blot analysis using anti-SAFV-3 antiserum.

To prepare Fc-tagged integrin beads, 293 T cells were transfected with pEFneo-ss-Fc, pEFneo-ITGαV-Fc, pEFneo-ITGβ8-Fc, or pEFneo-ITGβ3-Fc in the selected combinations using PEI Max. After 4 h of incubation, the medium was replaced with 2 ml of 1% FCS-DMEM per well. Seventy-two hours after transfection, the culture supernatants containing soluble Fc-tagged integrins were harvested. A total of 200 μl of the supernatant was incubated with 20 μl of protein A magnetic Dynabeads (10002D, Thermo Fisher Scientific) at 4 °C for 2 h with rotation. The integrin-Fc bead complexes were washed three times with 0.5% BSA-PBS (−). The binding of Fc-tagged integrin αVβ8 or

αVβ3 to the beads was confirmed by western blotting using a horseradish peroxidase-conjugated anti-mouse IgG antibody. For the pull-down of SAFV-3 by integrins αVβ8 and αVβ3, half of the prepared integrin-Fc bead complexes were suspended in 1 ml of 0.5% BSA-PBS, with or without 1.5 mM $CaCl_2$ and 1 mM $MgCl_2$ (PBS(−) or (+)), and mixed with SAFV-3 ($1 \times 10^6$ PFU). The mixture was then incubated at 4 °C for 2 h with rotation. After incubation, the beads were washed three times with 0.5% BSA-PBS (−) or 0.5% BSA-PBS (+) and once with PBS (−) or PBS (+). The precipitates were boiled in 2× SDS−PAGE sample loading buffer and subjected to western blot analysis using anti-SAFV-3 antiserum. ss-Fc beads were used as negative controls.

### Statistical analysis

Data were analyzed for statistical significance using one-way analysis of variance (ANOVA) with Dunnett's multiple comparison test and two-sided Welch's $t$-test. A $P$-value of <0.05 or <0.01 was considered statistically significant or highly significant. GraphPad Prism software, version 7 (GraphPad Software), was used for all statistical analyses.

### Reporting summary

Further information on research design is available in the Nature Portfolio Reporting Summary linked to this article.

## Data availability

The raw sequencing data from the CRISPR screens have been deposited in the DDBJ database (accession number PRJDB20430). The RNA sequences of the viral strains JPN08-404, JPN08-356, 987/Niigata/2007, and 1801-Yamagata-2009 have been deposited in GenBank or the DDBJ database under accession numbers HQ902242.1, LC865996, LC460463.1, and LC865997, respectively. Source data are provided with this paper.

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

## Acknowledgements

This work was supported by JSPS KAKENHI (Grant Number 21K07045, 24K10234 to T.O., 25K10386 to T.H.), AMED (Grant Number JP25fk0108716 to T.H.), Grant for Promoted Research from Kanazawa Medical University (S2023-4 to T.H.), Grant for Assistance KAKEN from Kanazawa Medical University (K2024-3 to T.H.), and a grant to M.H. from Yakult Honsha Co., Ltd. We thank Dr. Hiroyuki Shimizu from the National Institute of Infectious Diseases, Dr. Katsumi Mizuta from Yamagata Prefectural Institute of Public Health, and Dr. Chika Hirokawa from the Niigata Prefectural Institute of Public Health and Environmental Sciences for kindly providing the clinical isolates of SAFV. We thank Ms. Sumie Saito for technical assistance. We extend our special thanks to Dr. Yoshiro Ohara for his support as an observer.

## Author contributions

T.O. established the cell lines and performed the virological experiments. T.H. proposed this study, performed virological experiments, and wrote the manuscript. K.K. and N.N. performed genome-wide gene-KO screening. K.U. supported the various experiments. S.K. supervised the study design, performed virological experiments, and edited the final version of the manuscript. A.N. supported the analysis of viral binding to integrins using computer simulations. M.H. established the cell lines and was responsible for budget execution. All the authors have read and approved the final manuscript.

## Competing interests

The authors declare no competing interests.
