## [Transparent Peer Review file · Nature Communications]

Saffold Virus Exploits Integrin $\alpha V\beta 8$ and Sulfated Glycosaminoglycans as Cooperative Attachment Receptors for Infection.

Corresponding Author: Dr Toshiki HIMEDA

Version 0:

Reviewer comments:

Reviewer #1

(Remarks to the Author)

The manuscript investigates host factors essential for Saffold virus (SAFV) infection. Using a genome-wide CRISPR-Cas9 knockout screen, the authors identified sulfated glycosaminoglycans (GAGs) and integrin $\alpha V\beta 8$ as critical receptors facilitating SAFV binding via parallel pathways. Knockouts of genes required for GAG synthesis or integrin components (ITGAV and ITGB8) partially reduced viral susceptibility, while simultaneous knockouts conferred complete resistance in HeLa cells. Additionally, the expression of these integrins rendered cells susceptible to infection. These findings provide new insights into SAFV-host interactions. However, several conclusions require stronger supporting evidence, and improvements in the text and figure presentation are necessary for enhanced clarity and logic. Below are my suggestions for consideration:

Major points

1. Line 74-76: The authors conducted two independent screens but identified only one overlapping hit (SLC35B2) within the set thresholds. Could the authors clarify the reasons for this discrepancy? Is it due to differences in library coverage, screening conditions, or other factors?
2. Related to the first point, one of the screens identifies MYADM, a known parechovirus receptor, and SETD3, a methyltransferase critical for enterovirus infection. They are absent in the other screen. Could these "hits" be a cross-contamination of other screens run in parallel using other picornaviruses, or are these biologically relevant? To enhance rigor, the validation of these additional hits from the screens is strongly recommended. It could exclude potential artefacts or provide valuable insights into shared host factors among viruses in the same family.
3. The authors should deposit raw sequencing data on a public platform from the CRISPR screens described in the manuscript, and include a comprehensive list of results as supplementary information in the form of an excel file that contains q-values and log fold changes per gene.
4. The authors demonstrated that virus binding was significantly reduced in HeLaN- Δ SLC cells but not in HeLaN- Δ B8 cells. What is the underlying cause of reduced virus infection in HeLaN- Δ B8 cells? Could it be due to impaired internalization or another step in the viral entry process? Additionally, would virus binding decrease in HeLaN- Δ AVB8 cells? Conversely, does virus attachment increase in HeLaN- Δ SLC+human B8 cells, given that expressing B6 alone in BHK21 cells already significantly enhances virus infection? Providing data to clarify these mechanisms would strengthen the conclusions.
5. The authors used an RGD peptide to mask the RGD-binding site on AVB8, effectively blocking SAFV-3 infection. This led to the conclusion that SAFV-3 interacts with integrin AVB8 via the RGD-binding site. However, SAFV-3 contains an RAD sequence, and SAFV-2 contains an RLD sequence, that are located in different viral capsid proteins. What is the hypothesis underlying these statements? Are these sequences potential integrin-binding sites for the viruses? Additionally, could RAD or RLD peptides block virus infection? Do the authors suggest that SAFV-2 and SAFV-3 interact with integrins differently? Alternatively, is it possible that the RGD peptide induces conformational changes in integrins, thereby inhibiting SAFV infection? Clarification and supporting data would enhance the understanding of this mechanism.
6. Line 222-224: The authors should specify the mutants derived from latent-TGF- β binding, along with the rationale behind their use in virus binding studies. This additional detail would help readers better understand the experimental design and conclusions, allowing for a clearer and more independent interpretation of the manuscript.

Minor points

1. Fig.1a: Clarify in the legend that the two panels represent independent screens. Also, include threshold lines (e.g., q-value < 0.01, log₂FC > 4) directly on Figures 1a and 2a to improve their interpretability.

7. Line 84: Could the authors add a description and explain why EXT1 was chosen for validation instead of EXT2, which was identified in the screen?
8. Line 104: The authors performed secondary screens to identify additional SAFV-3 receptors using HeLaN-ΔSLC cells. While it is reasonable that genes involved in GAG synthesis were excluded, it remains unclear why none of the other hits from the primary screens were detected. It seems unlikely that all these hits are related to GAG synthesis.
9. The authors showed that knocking out either SLC35B2 or integrin alone retained some susceptibility to SAFV-3 in HeLa-N cells, while dual knockouts of SLC35B2 and ITGAV or ITGB8 led to complete resistance. Have the authors considered testing the double knockout of ITGAV and ITGB8?
10. Line 148-150: The non-detection of endogenous $\beta 8$ integrin may stem from low antibody sensitivity rather than its absence. This should be clarified, as antibody sensitivity affects the reliability of protein detection.
11. Add protein ladders to western blots in figures such as 3a, 4c, and 5f to ensure proper interpretation.
12. Consider merging Figures 4c and 4d into Figure 3 to clarify that $\beta 8$ integrin overexpression increases susceptibility without a species barrier. Currently, Figure 3 could be misleading.
13. Fig.4b: Include western blots showing integrin expression levels to validate the results.
14. Ensure figure legends are self-explanatory, including abbreviations like SAFV-Ag, etc.
15. Line 207-208: Provide a brief explanation of the role of $\text{Ca}^{2+}/\text{Mg}^{2+}$ in the immunoprecipitation experiments.
16. Extended Data Fig. 8: The suggestions in the figure and figure legend that "a portion of viruses bound to sulfated GAGs may subsequently interact with integrin $\alpha V\beta 8$ " and that integrins assist in virus uncoating require additional experimental evidence to support these claims.

Reviewer #2

(Remarks to the Author)

This study identifies sulfated glycosaminoglycans (GAGs) and integrin $\alpha V\beta 8$ as key receptors for Saffold virus (SAFV) infection using a genome-wide CRISPR-Cas9 knockout screen. The findings suggest that these molecules function in parallel pathways to mediate viral entry. However, several aspects of the manuscript lack rigor, and additional experiments are necessary to fully justify the proposed conclusions and strengthen the mechanistic understanding of SAFV-host interactions. These concerns highlight the need for a more rigorous approach to data analysis, presentation, and validation to strengthen the manuscript's conclusions. I have detailed these specific aspects below:

1. CRISPR Screen Analysis: The selection of hits from the screen appears to be manually curated rather than analyzed with established CRISPR screen analysis tools such as MAGECK, HiTSelect, or BAGEL. Using such tools would provide a more robust and statistically sound assessment of gene hits.
A more rigorous analysis is necessary to explain discrepancies, such as why integrin subunit genes identified in Figure 2 were not identified in Figure 1. A review of these various tools is PMID: 36937794
2. Issues in Data Presentation:
 - 2a. Bar Graphs: Should display individual data points to improve transparency.
The number of independent experiments should be clearly indicated.
 - 2b. Graph Axes and Detection Limits: Axes should start at zero instead of being cropped (e.g., Figure 1D).
 - 2c. Limits of detection should be included in relevant graphs.
3. Representative Images and Experiment Replication: Crystal-violet-stained plates (e.g., Figure 6) should indicate whether the image is representative of multiple experiments or if the experiment was only performed once.
4. Need for Additional Validation Experiments: Specificity of Knockout Resistance: Knockout (KO) cells should be infected with another picornavirus with a known receptor to confirm that the resistance observed is specific to SAFVs and not due to a general inability of the KO cells to support viral replication.
5. Species-Specificity Claims: The claim that differences in integrin $\alpha V\beta 8$ do not determine the species specificity of SAFV-3 infection (lines 181-182) is only supported by data from mouse and hamster. Additional species should be included to substantiate this conclusion.
6. Clarification of Graphical Abstract: The graphical abstract implies an interdependent relationship between the sulfated GAG and integrin $\alpha V\beta 8$ pathways. However, the manuscript does not provide direct evidence supporting interdependence. The authors should clarify whether they believe the pathways are independent or interconnected and provide experimental support for their proposed model.

Reviewer #3

(Remarks to the Author)

In this work, Okuwa et al. aim to identify Saffold virus (SAFV) receptors. SAFV is a relatively recently identified human virus in the *Cardiovirus* genus. SAFV has been associated with gastrointestinal and respiratory illnesses. Similar to other picornaviruses, such as Enterovirus A71 (EV-A71), SAFV has been suggested to play a role in hand, foot, and mouth disease as well as encephalitis and meningitis. Little is known about SAFV pathogenesis and given the high seroprevalence, a better understanding is warranted. Therefore, the authors used a genome-wide CRISPR-Cas9 knockout screening to identify potential receptors involved in SAFV infection. Through this screen, the authors first identify that two proteins (SLC35B2 and EXT-2) involved in sulfated GAG synthesis and host sulfation are critical for SAFV infection.

Subsequently, the authors also identify that ITGAV and ITGB8, two integrin genes, are also important for infection in the absence of GAG synthesis. Through double knockouts and various confirmation experiments, the authors propose that SAFV entry involves two parallel pathways involving sulfated GAGs and integrin α V β 8. The experiments presented in this paper are thorough and provide important insights into the entry steps of SAFV. Overall, I am convinced that the paper has merit for publication and will be an important milestone in our understanding of SAFV. Nevertheless, a few points below require attention and may further improve the manuscript.

- 1) Figure 1A and the text on line 81 state that EXT2 was identified in the CRISPR screen. However, the authors proceed with EXT1 knockout (KO, line 84) and present results using this KO. The rationale for the switch from EXT-2 to EXT-1 has not been described. Why have the authors not performed further experiments with EXT2? Further, please replace EXT-2 with EXT2 for consistency.
- 2) Further, EXT2 was only identified on the second screen and not the first. Similarly, other genes are different between the two screens. Why focus on EXT2?
- 3) In general, their CRISPR screens seem to identify very few host factors. However, in our experience, in addition to entry factors, these screens usually also pick on other critical host factors that are involved in downstream aspects of viral replication, other than viral entry. Can the authors comment on the lack of these genes in their screens?
- 4) Identification of SLC35B2 is interesting as this has been identified as a critical factor for EV-A71 infection (PMID: 35420441). In that manuscript, Guo et al demonstrate that SLC35B2 does not only play a role in heparan sulfate sulfation but also is important for regulating the overall protein tyrosine sulfation. It would be useful to see the sulfation status of the other proteins identified in the CRISPR screens. In particular, MYADM could be a post-attachment target (Factor X) as it was also identified as an essential entry factor for human parechoviruses (PMID: 38658526, 37002207). Please discuss the role of host sulfation and cite this paper.
- 5) Although the authors use clinical isolates in the later experiments, it is not clear how many passages have been performed. It is necessary to show by sequencing that these passaged clinical isolates do not have mutations in the receptor binding region compared to the original clinical strain. Picornaviruses are known to pick up heparan sulfate binding mutations within a few passages and this information is necessary to rule out if the HS-binding phenotype is a culture artefact.
- 6) On that note, the role of HS-binding in viral infection is a topic of debate (PMID: 31266258) and must be discussed.
- 7) Furthermore, although HS has been suggested to be important for other picornavirus (EV-A71, EV-D68) infections using cell lines, their role when using primary human cell models is not as clear. Similarly, integrins were shown to play a role in parechovirus infection in cell lines but did not influence their infection in human airway epithelial cultures. Thus, the use of only cell lines in this study must be discussed.
- 8) In Figure 5a, the B8 KO does not have an effect on viral binding but the soluble integrin in 1C has an effect at the highest concentration. Can you comment on this?

Reviewer #4

(Remarks to the Author)

Version 2:

Reviewer comments:

Reviewer #1

(Remarks to the Author)

The authors have satisfactorily addressed my points in their revised manuscript.

Reviewer #3

(Remarks to the Author)

The authors have addressed my comments and those of the other reviewers sufficiently. Having rerun the CRISPR screens and performing additional experiments demonstrates clearly the role of sulfated GAGs and Integrin α V β 8 in viral entry. The only suggestion I have is to change the title from "dual receptors" to "dual attachment receptors", as they have not explored entry and uncoating in this set of experiments.

Reviewer #4

(Remarks to the Author)

We sincerely thank the reviewers and editors for their insightful and constructive comments on our original manuscript. This response addresses all concerns raised and outlines the corresponding modifications made to the text and figures. Reviewer comments are presented in *italics*, and our responses are provided in **bold**.

Reviewer #1 (Remarks to the Author):

Major points

1. Line 74-76: The authors conducted two independent screens but identified only one overlapping hit (SLC35B2) within the set thresholds. Could the authors clarify the reasons for this discrepancy? Is it due to differences in library coverage, screening conditions, or other factors?

Reviewer 2 also raised a similar point and recommended improving the analysis method. In response, we reanalyzed the data using the MAGeCK algorithm. To enhance clarity, we have also included a diagram highlighting overlapping genes identified in the two independent screens (revised version, Lines 73-83; Fig. 1a and Fig. 2a). As a result, SLC35B2, EXT1, and PA2G4 emerged as the primary overlapping genes.

The limited number of overlapping hits is likely due to the nature of the primary screen: SAFV infection in HeLa-N cells appears to utilize multiple entry and replication pathways, making it unlikely that a single gene knockout would confer strong resistance. Indeed, the number of surviving cells post-infection in the primary screen was extremely low (revised version, Lines 78-80). Nevertheless, reanalysis identified PA2G4—previously reported as important for the FMDV replication cycle—as an overlapping hit, suggesting that the screening process was functioning as expected.

2. Related to the first point, one of the screens identifies MYADM, a known parechovirus receptor, and SETD3, a methyltransferase critical for enterovirus infection. They are absent in the other screen. Could these “hits” be a cross-contamination of other screens run in parallel using other picornaviruses, or are these biologically relevant? To enhance rigor, the validation of these additional hits from the screens is strongly recommended. It could exclude potential artefacts or provide valuable insights into shared host factors among viruses in the same family.

As the reviewer pointed out, the first screen using the JPN08-404 strain was conducted in parallel with screens for parechovirus A3 and coxsackievirus. Given the nature of the analytical equipment used, we cannot rule out the possibility of cross-contamination. To address this concern, we performed validation experiments for MYADM and SETD3 and confirmed that

they were likely false positives, probably due to cross-contamination (shown in the figure below).

MYADM and SETD3 were not identified as overlapping genes in the two independent screens and are therefore not mentioned in the manuscript.

In addition, we conducted validation experiments for other candidate genes identified in both independent screens, and the results have been included in the revised manuscript (revised version, Lines 103-108; Supplementary Fig. 3a).

3. The authors should deposit raw sequencing data on a public platform from the CRISPR screens described in the manuscript, and include a comprehensive list of results as supplementary information in the form of an excel file that contains q-values and log fold changes per gene.

As suggested, we have deposited the raw sequencing data from the CRISPR screens in the DDBJ database under the accession number [PRJDB20430], and made it publicly accessible. In addition, a comprehensive list of screening results, including q-values and log fold changes for each gene, has been provided as Source data file in Excel format.

4. The authors demonstrated that virus binding was significantly reduced in HeLaN-ΔSLC cells but not in HeLaN-ΔB8 cells. What is the underlying cause of reduced virus infection in HeLaN-ΔB8 cells? Could it be due to impaired internalization or another step in the viral entry process?

We believe that the reduced viral infection observed in HeLaN-ΔB8 cells is due to the loss of integrin α V β 8-mediated infection. The absence of a decrease in virus binding in these cells is likely attributable to the limited sensitivity of our detection system, which may not be sufficient to measure virus binding at the low endogenous levels of integrin β 8 (revised Fig. 6a-c). Supporting this interpretation, overexpression of integrin α V β 8 led to a marked increase in virus binding (revised Fig. 6a) and a corresponding enhancement in viral susceptibility (revised Fig. 6b).

Alternatively, it is possible that sulfated GAGs serve as primary attachment receptors in a cooperative pathway and assist integrin $\alpha V\beta 8$ mediated infection. In this scenario, HeLaN- $\Delta B8$ cells, which retain functional sulfated GAGs, would exhibit reduced viral infection but unchanged binding levels—consistent with our observations. We have incorporated these explanations into the revised manuscript (Lines 217-232 and 333-339).

Although our findings support the involvement of integrin $\alpha V\beta 8$ in the infection process, the precise mechanism -whether it facilitates viral internalization, uncoating, or another step-remains unclear. Therefore, we cannot yet provide a definitive explanation for the molecular basis of reduced infection in HeLaN- $\Delta B8$ cells.

Additionally, would virus binding decrease in HeLaN- $\Delta AVB8$ cells? Conversely, does virus attachment increase in HeLaN- ΔSLC +human B8 cells, given that expressing B6 alone in BHK21 cells already significantly enhances virus infection? Providing data to clarify these mechanisms would strengthen the conclusions.

Integrin $\alpha V\beta 8$ is expressed on the cell surface only as a heterodimer (ref. 28). Therefore, knockout of either subunit is expected to decrease the surface expression of the both subunits. Consequently, additional knockout of integrin αV in HeLaN- $\Delta B8$ cells is unlikely to further reduce virus binding. To underscore the importance of integrin heterodimer formation, we have cited an additional reference (revised version, ref. 28) and included new FACS data showing that knockout of integrin αV leads to a reduction in cell surface expression of $\beta 8$ (revised version, Lines 120-124; Supplementary Fig. 5).

BHK cells endogenously express integrin αV but not $\beta 8$. Upon overexpression of $\beta 8$ alone, these cells gained viral susceptibility, likely due to the formation of functional $\alpha V\beta 8$ heterodimers, which may also enhance virus binding. Similarly, overexpression of integrin $\beta 8$ in HeLaN- ΔSLC cells could potentially increase virus binding to some extent, by forming additional heterodimers with the available αV . However, we believe such experiments would not provide significant new insights into our conclusions, and therefore we opted not to perform them.

5. The authors used an RGD peptide to mask the RGD-binding site on AVB8, effectively blocking SAFV-3 infection. This led to the conclusion that SAFV-3 interacts with integrin AVB8 via the RGD-binding site. However, SAFV-3 contains an RAD sequence, and SAFV-2 contains an RLD sequence, that are located in different viral capsid proteins. What is the hypothesis underlying these statements? Are these sequences potential integrin-binding sites for the viruses? Additionally, could RAD or RLD peptides block virus infection? Do the authors suggest that SAFV-2 and SAFV-3 interact with integrins

differently? Alternatively, is it possible that the RGD peptide induces conformational changes in integrins, thereby inhibiting SAFV infection? Clarification and supporting data would enhance the understanding of this mechanism.

To test our hypothesis, we conducted additional experiments using RAD and RLD peptides to assess their ability to block virus infection. We also generated mutant viruses with point mutations introduced into the corresponding capsid regions to evaluate whether these sequences function as integrin-binding sites. These analyses revealed that neither the RAD nor RLD motifs serve as the integrin-binding site, thus disproving our initial hypothesis.

Our findings now support the conclusion that SAFV interacts with the RGD-binding pocket of integrins; however, the viral binding motif is not an RGD-like sequence but likely an alternative, previously unidentified sequence. These new data have been incorporated into the revised manuscript (revised version: Lines 253-283 and 346-359; Fig. 7).

Furthermore, as the reviewer insightfully pointed out, it is plausible that binding of the RGD peptide or mutations in the integrin-binding pocket could induce conformational changes that influence virus interaction. We agree that structural studies will be necessary to fully understand the molecular details of SAFV–integrin interactions, and we have added a statement to this effect in the revised text (Lines 358-359).

6. Line 222-224: The authors should specify the mutants derived from latent-TGF- β binding, along with the rationale behind their use in virus binding studies. This additional detail would help readers better understand the experimental design and conclusions, allowing for a clearer and more independent interpretation of the manuscript.

As mentioned in response to comment 5, we have added new experiments and significantly revised this section of the manuscript. In accordance with the reviewer's suggestion, we have provided a more detailed explanation regarding the mutants derived from latent-TGF- β binding and the rationale behind their use in virus binding studies. This additional information has been included in the revised manuscript (revised version: Lines 254-258, Fig. 7).

Minor points

1. Fig.1a: Clarify in the legend that the two panels represent independent screens. Also, include threshold lines (e.g., $q\text{-value} < 0.01$, $\log_2FC > 4$) directly on Figures 1a and 2a to improve their interpretability.

To enhance clarity, we have updated the figure and legend to indicate that Fig.1a represents the

combined result of two independent screens. As requested, we have included threshold lines (e.g., $-\log_{10}(\text{RRA score}) > 2$) directly on Figures 1a and 2a. Additionally, overlapping genes that exceed the threshold are now highlighted in color to improve the interpretability of the figures (revised version: Fig. 1a and 2a).

7. Line 84: Could the authors add a description and explain why EXT1 was chosen for validation instead of EXT2, which was identified in the screen?

As stated in response to comment 5, we have revised both the analysis and display methods to improve clarity. As a result, SLC35B2, EXT1, and PA2G4 were identified as the main overlapping genes. We believe these changes have addressed the confusion regarding the choice of EXT1 for validation. (Revised version: Lines 73-83, Fig. 1a)

8. Line 104: The authors performed secondary screens to identify additional SAFV-3 receptors using HeLaN- Δ SLC cells. While it is reasonable that genes involved in GAG synthesis were excluded, it remains unclear why none of the other hits from the primary screens were detected. It seems unlikely that all these hits are related to GAG synthesis.

As with the first screening, we have revised the analysis and display methods for the secondary screens (Revised version: Fig. 2a). PA2G4 was the only validated hit, besides ITGAV and ITGB8. The identification of PA2G4 as an overlapping hit suggests that the screening functioned properly. The relatively low number of hit genes is likely due to the existence of alternative molecules involved in the SAFV replication cycle. We hypothesize that knocking out a single gene may not result in resistant cells because multiple pathways may contribute to viral replication. This explanation has been added to the revised manuscript. (Revised version: Lines 153-156, and 315-321)

9. The authors showed that knocking out either SLC35B2 or integrin alone retained some susceptibility to SAFV-3 in HeLa-N cells, while dual knockouts of SLC35B2 and ITGAV or ITGB8 led to complete resistance. Have the authors considered testing the double knockout of ITGAV and ITGB8?

As stated in response to comment 4, integrin α V β 8 is expressed on the cell surface only as a heterodimer (Revised version: Ref. 28). Therefore, a double knockout of both integrin α V and β 8 would not provide additional insights, as these subunits are required for the formation of functional integrin α V β 8 heterodimers.

10. Line 148-150: *The non-detection of endogenous $\beta 8$ integrin may stem from low antibody sensitivity rather than its absence. This should be clarified, as antibody sensitivity affects the reliability of protein detection.*

As the reviewer suggested, the sensitivity of the antibody may be low, as indicated by the signal for exogenous hamster integrin $\beta 8$. However, the observation that BHK-21 cells acquire SAFV-3 susceptibility only after the exogenous expression of hamster $\beta 8$ suggests that BHK-21 cells do not express integrin $\beta 8$ at a level sufficient for SAFV infection.

11. *Add protein ladders to western blots in figures such as 3a, 4c, and 5f to ensure proper interpretation.*

As the reviewer suggested, we have added protein ladders to the western blots in the revised version for better interpretation. (revised ver. Fig. 4a, 4e and 7a)

12. *Consider merging Figures 4c and 4d into Figure 3 to clarify that $\beta 8$ integrin overexpression increases susceptibility without a species barrier. Currently, Figure 3 could be misleading.*

As the reviewer suggested, we have merged Figures 4c and 4d into Figure 3 in the revised version to clarify that $\beta 8$ integrin overexpression increases susceptibility without a species barrier. (revised ver. Lines 195-200, Fig. 4e and 4f)

13. *Fig.4b: Include western blots showing integrin expression levels to validate the results.*

Verification of integrin expression levels was performed by FACS analysis. Please refer to Figure 5a in the revised version for the data.

14. *Ensure figure legends are self-explanatory, including abbreviations like SAFV-Ag, etc.*

As the reviewer suggested, we have added the missing abbreviation explanations to the figure legends. Please refer to the revised version of Figure 1 and 6.

15. Line 207-208: *Provide a brief explanation of the role of Ca^{2+} / Mg^{2+} in the immunoprecipitation experiments.*

Integrins undergo a Ca^{2+} / Mg^{2+} -dependent conformational change to their active form, which enables ligand binding to the RGD-binding pocket. Therefore, pull-down assays were performed

both in the presence and absence of Ca^{2+} / Mg^{2+} . This explanation has been added to the revised version. (revised ver. Lines 242-245, and 249-250)

16. Extended Data Fig. 8: The suggestions in the figure and figure legend that “a portion of viruses bound to sulfated GAGs may subsequently interact with integrin $\alpha\text{V}\beta 8$ ” and that integrins assist in virus uncoating require additional experimental evidence to support these claims.

Experimental evidence supporting the statement "a portion of viruses bound to sulfated GAGs may subsequently interact with integrin $\alpha\text{V}\beta 8$ " was shown in Extended Data Fig. 5 and lines 162-165 of the original text. However, since this presentation was not sufficient, we added the results of the adsorption analysis and included them in the main text as regular figures. (revised ver. Lines 286-301, and Fig. 8)

It is unclear whether integrins assist in viral uncoating, and we believe that elucidating this question is a future research direction that is beyond the scope of this manuscript. As pointed out by the reviewer, the original graphical abstract was misleading. Therefore, in the revised version, we removed any content that could suggest uncoating. To improve clarity, we have moved this figure from an extended figure to a regular figure. (revised ver. Fig. 9)

Reviewer #2 (Remarks to the Author):

1. CRISPR Screen Analysis: The selection of hits from the screen appears to be manually curated rather than analyzed with established CRISPR screen analysis tools such as MAGeCK, HiTSelect, or BAGEL. Using such tools would provide a more robust and statistically sound assessment of gene hits. A more rigorous analysis is necessary to explain discrepancies, such as why integrin subunit genes identified in Figure 2 were not identified in Figure 1. A review of these various tools is PMID: 36937794

Reviewer 1 also raised a similar concern and suggested improving the analysis method. In response to this feedback, we reanalyzed the data using the MAGeCK method. To improve clarity, we have also added a diagram highlighting the overlapping genes between the two screening results (revised ver. Fig. 1a and 2a).

The discrepancy in identifying integrin subunit genes between Figures 1 and 2 is likely due to the fact that, in the primary screening involving the combination of SAFV and HeLa-N cells, multiple infection pathways exist, and knocking out a single gene did not result in resistant cells. Notably, the number of surviving cells after SAFV infection was extremely low in the primary screening (revised ver. Lines 78-80). Upon reanalysis, we found that PA2G4, which is known to be important in the FMDV replication cycle, was an overlapping hit, further confirming that the screening method functioned properly.

2. Issues in Data Presentation:

2a. Bar Graphs: Should display individual data points to improve transparency. The number of independent experiments should be clearly indicated.

As suggested, we have revised the bar graphs to display individual data points for better transparency. Additionally, the number of independent experiments is now clearly indicated in the revised figures (revised ver. Fig. 4c, 6a, 6d, 6e, 7a, 7c, 8b, 8c, and 8d).

2b. Graph Axes and Detection Limits: Axes should start at zero instead of being cropped (e.g., Figure 1D).

2c. Limits of detection should be included in relevant graphs.

As suggested, we have revised the graphs to ensure that the axes start at zero instead of being cropped. Additionally, we have included the limits of detection in the relevant graphs (revised ver. Fig. 1d, 2d, 4d and 7a).

3. *Representative Images and Experiment Replication: Crystal-violet-stained plates (e.g., Figure 6) should indicate whether the image is representative of multiple experiments or if the experiment was only performed once.*

As suggested, we have added a statement to the figure legends indicating that the images shown are representative of multiple experiments (revised ver. Figure Legends 1-8).

4. *Need for Additional Validation Experiments: Specificity of Knockout Resistance: Knockout (KO) cells should be infected with another picornavirus with a known receptor to confirm that the resistance observed is specific to SAFVs and not due to a general inability of the KO cells to support viral replication.*

As the reviewer suggested, we infected each KO cell line with Encephalomyocarditis virus and Coxsackievirus B3 and confirmed that these cells retained the ability to support the replication of viruses with known receptors. The results have been added to the revised version (revised ver. Lines 137-141, Supplementary Fig. 6).

5. *Species-Specificity Claims: The claim that differences in integrin $\alpha V\beta 8$ do not determine the species specificity of SAFV-3 infection (lines 181-182) is only supported by data from mouse and hamster. Additional species should be included to substantiate this conclusion.*

As the reviewer suggested, the claim regarding species specificity may have been an overstatement. Since the concept of species specificity is not the central focus of this manuscript, we have revised the wording to: "Integrin $\beta 8$ is compatible between humans and rodents in SAFV infection." This revision is reflected in the revised version (revised ver. Lines 195-200, 330-332).

6. *Clarification of Graphical Abstract: The graphical abstract implies an interdependent relationship between the sulfated GAG and integrin $\alpha V\beta 8$ pathways. However, the manuscript does not provide direct evidence supporting interdependence. The authors should clarify whether they believe the pathways are independent or interconnected and provide experimental support for their proposed model.*

Experimental evidence supporting the interdependent relationship between the sulfated GAG and integrin $\alpha V\beta 8$ pathways was shown in Extended Data Fig. 5 and lines 162-165 of the original version. However, this presentation was not sufficiently clear. In the revised version, we have

added the results of the adsorption analysis and included them as regular figures for better clarity (revised ver. Lines 286-301, Fig. 8).

To further clarify whether we consider the pathways to be independent or interconnected, we modified the phrase 'Parallel pathway' to 'dual and cooperative pathway'. This revision has been incorporated into the title, abstract, graphical abstract, and relevant sections throughout the manuscript (revised ver. Lines 3, 37-38, 65-67, 144-145, 159-162, 229-232, 300-301, 385-386 and Fig. 9).

Reviewer #3 (Remarks to the Author):

1) Figure 1A and the text on line 81 state that EXT2 was identified in the CRISPR screen. However, the authors proceed with EXT1 knockout (KO, line 84) and present results using this KO. The rationale for the switch from EXT-2 to EXT-1 has not been described. Why have the authors not performed further experiments with EXT2? Further, please replace EXT-2 with EXT2 for consistency.

Following the suggestions of Reviewer 1 and Reviewer 2, we reanalyzed the data using more conventional analytical methods. As a result, EXT1, but not EXT2, was identified as an overlapping hit in the two screenings. We apologize for the confusion caused by the initial choice of analysis method and presentation in the original manuscript. This reanalysis has resolved the issue, and we have updated the manuscript accordingly. Additionally, we have corrected the term "EXT-2" to "EXT2" for consistency throughout the manuscript. (revised ver. Lines 73-83, Fig. 1a)

2) Further, EXT2 was only identified on the second screen and not the first. Similarly, other genes are different between the two screens. Why focus on EXT2?

Please refer to our response to Comment 1 above. We reanalyzed the data, and as a result, EXT1, but not EXT2, was identified as an overlapping hit in both screenings. We have clarified the rationale for focusing on EXT1 in the revised manuscript.

3) In general, their CRISPR screens seem to identify very few host factors. However, in our experience, in addition to entry factors, these screens usually also pick on other critical host factors that are involved in downstream aspects of viral replication, other than viral entry. Can the authors comment on the lack of these genes in their screens?

The limited number of hit genes identified in our CRISPR screens is likely due to the fact that, in the case of the SAFV and HeLa-N combination, multiple infection pathways and alternative molecules for replication exist. As a result, knockout of a single gene did not lead to complete resistance. However, reanalysis revealed that PA2G4 was identified as an overlapping hit in all screenings, indicating that the screening process was functioning properly. We have added this explanation to the revised manuscript. (revised ver. Lines 103-108, 153-156 and 315-321)

4) Identification of SLC35B2 is interesting as this has been identified as a critical factor for EV-A71 infection (PMID: 35420441). In that manuscript, Guo et al demonstrate that SLC35B2 does not only

play a role in heparan sulfate sulfation but also is important for regulating the overall protein tyrosine sulfation. It would be useful to see the sulfation status of the other proteins identified in the CRISPR screens. In particular, MYADM could be a post-attachment target (Factor X) as it was also identified as an essential entry factor for human parechoviruses (PMID: 38658526, 37002207). Please discuss the role of host sulfation and cite this paper.

As the reviewers suggested, we also recognized the importance of examining overall protein tyrosine sulfation, a role attributed to SLC35B2. To investigate this, we compared the reduced SAFV susceptibility in EXT1 KO and SLC35B2 KO HeLa-N cells. Our results showed no major difference, suggesting that the reduced SAFV sensitivity observed in SLC35B2 KO cells was primarily due to the loss of sulfated GAGs, with the overall protein tyrosine sulfation playing a minor role (original ver. Fig. 1c and lines 87-95). To emphasize this point, we have cited the references (revised ver. ref. 22-25) and revised the relevant sentences in the text (revised ver. Lines 94-97 and 360-364).

Regarding MYADM, we performed the first screening with JPN08-404 in parallel with screenings for parechovirus A3 and Coxsackievirus. Due to the nature of the analytical equipment used, the possibility of cross-contamination cannot be ruled out. To address this, we conducted a verification experiment for MYADM and SETD3, confirming that these were false positives, likely caused by cross-contamination. Please refer to our response to Comment 2 from Reviewer 1 for more details.

5) Although the authors use clinical isolates in the later experiments, it is not clear how many passages have been performed. It is necessary to show by sequencing that these passaged clinical isolates do not have mutations in the receptor binding region compared to the original clinical strain. Picornaviruses are known to pick up heparan sulfate binding mutations within a few passages and this information is necessary to rule out if the HS-binding phenotype is a culture artefact.

6) On that note, the role of HS-binding in viral infection is a topic of debate (PMID: 31266258) and must be discussed.

The passage number for each clinical strain used in our experiments has been added as Supplementary Table 1. The sequence data for these clinical strains were deposited in the DDBJ database and assigned accession numbers. However, since the sequence data were obtained after the clinical strains were received, we cannot guarantee that the sequences of the original, unpassaged clinical strains were fully preserved.

In response to comment 6, we also added a discussion regarding the possibility that the HS-binding phenotype may be influenced by cell culture adaptation. This potential

consideration has been included in the revised version (revised ver. Lines 162-163, 373-384, Supplementary Table 1).

7) Furthermore, although HS has been suggested to be important for other picornavirus (EV-A71, EV-D68) infections using cell lines, their role when using primary human cell models is not as clear. Similarly, integrins were shown to play a role in parechovirus infection in cell lines but did not influence their infection in human airway epithelial cultures. Thus, the use of only cell lines in this study must be discussed.

As this study was limited to analysis using cultured cell lines, it remains unclear how closely the results correlate with in vivo infection. We acknowledge that the role of HS and integrins may differ when using primary human cell models, as seen in previous studies on other picornaviruses. It is our hope that future analyses, including those using primary human cell models, will provide further insights. These points have been added to the discussion section in the revised version (revised ver. Lines 380-384).

8) In Figure 5a, the B8 KO does not have an effect on viral binding but the soluble integrin in IC has an effect at the highest concentration. Can you comment on this?

The lack of a reduction in virus binding in HeLaN- Δ B8 cells is likely due to the insufficient sensitivity of our detection system, which may have prevented us from detecting virus binding at the low levels of β 8 expression in WT-HeLaN cells (revised Fig. 6a-c). Supporting this interpretation, the overexpression of integrins, which helped compensate for the low detection sensitivity, resulted in a significant increase in virus binding (revised Fig. 6a) and enhanced virus susceptibility (revised Fig. 6b). To improve clarity for readers, we have added an explanation in the revised text (revised ver. Lines 217-232, and 333-339).

In contrast, in the competition assay with soluble integrins, HeLaN- Δ SLC+human AVB8 cells were used to clearly demonstrate the effect.

Reviewer #4 (Remarks to the Author):

We thank you for participating in the peer review of our manuscript.

We sincerely thank the reviewers and editors for their careful reading and evaluation of our revised manuscript. We are pleased that the reviewers found our revisions, including the newly performed CRISPR screens and additional experiments, to be satisfactory, and that these clearly demonstrate the role of sulfated GAGs and integrin $\alpha V\beta 8$ in viral entry. We believe that all major concerns have been thoroughly addressed. Reviewer comments are presented in *italics*, and our responses are provided in **bold**.

Reviewer #3 (Remarks to the Author):

The authors have addressed my comments and those of the other reviewers sufficiently. Having rerun the CRISPR screens and performing additional experiments demonstrates clearly the role of sulfated GAGs and Integrin $\alpha V\beta 8$ in viral entry. The only suggestion I have is to change the title from "dual receptors" to "dual attachment receptors", as they have not explored entry and uncoating in this set of experiments.

We thank Reviewer #3 for acknowledging the sufficiency of our revisions and the clarity provided by the new experimental data, which strongly supports the roles of sulfated GAGs and integrin $\alpha V\beta 8$.

We agree with the suggestion to refine the term "dual receptors" in the title to "dual attachment receptors" to more accurately reflect the scope of our current study, which primarily focuses on the attachment phase and does not extensively examine the subsequent steps of entry and uncoating. We have revised the manuscript title based on your suggestion and the editorial office's requirement to: "Saffold Virus Exploits Integrin $\alpha V\beta 8$ and Sulfated Glycosaminoglycans as Cooperative Attachment Receptors for Infection"